# FPGA Implementation of Parameter-Switching Scheme to Stabilize Chaos in Fractional Spherical Systems and Usage in Secure Image Transmission

**Vincent-Ademola Adeyemi** [1] , **Esteban Tlelo-Cuautle** [2] , **Yuma Sandoval-Ibarra** [3]
**and Jose-Cruz Nuñez-Perez** [1,*]

1   Instituto Politécnico Nacional, IPN-CITEDI, Tijuana 22435, Mexico; vademola@citedi.mx
2   Instituto Nacional de Astrofísica, Óptica y Electrónica, INAOE, San Andres Cholula 72840, Mexico; etlelo@inaoep.mx
3   Departamento de Posgrado, Universidad Politécnica de Lázaro Cárdenas, UPLC, Lázaro Cárdenas 60950, Mexico; yumasandoval@uplc.edu.mx
*   Correspondence: jnunez@ipn.mx or nunez@citedi.mx; Tel.: +52-55-5729-6000

**Abstract:** The main objective of this work was to implement the parameter-switching chaos control scheme for fractional-order spherical systems and develop a chaos-based image encryption and transmission system. The novelty in the developed secure communication system is the application of the parameter-switching scheme in the decryption of RGB and grayscale images, which undergo one round of encryption using the chaotic states of the fractional system and a diffusion process. The secure communication system has a synchronized master and slave topology, resulting in transmitter and receiver systems for encrypting and decrypting images, respectively. This work was demonstrated numerically and also implemented on two FPGAs, namely Artix-7 AC701 and Cyclone V. The results show that the parameter-switching scheme controls chaos in the fractional-order spherical systems effectively. Furthermore, the performance analysis of the image encryption and transmission system shows that there is no similarity between the original and encrypted images, while the decryption of the encrypted images is without a loss of quality. The best result in terms of the encryption was obtained from the chaotic state $x$ of the fractional-order system, with correlation coefficients of 0.0511 and 0.0392 for the RGB and grayscale images, respectively. Finally, the utilization of the FPGA logical resources shows that the implementation on Artix-7 AC701 is more logic-efficient than on Cyclone V.

**Keywords:** chaos; encryption; fractional order; FPGA; image transmission; parameter switching; synchronization





## 1. Introduction

The fundamental theory of fractional calculus dates back to over 300 years ago [1,2]. Gottfried Wilhelm Leibniz, a German mathematician, invented the notation $(d^n y)/(dx^n)$ for the derivatives of integer order, where order $n$ is a non-negative integer. Generally, fractional calculus allows for the integration and differentiation of any positive real order—fractional, complex, or irrational. Since the birth of fractional calculus, it has continued to have major impacts in science and engineering, where it is applied to the modeling of complex and dynamic systems, which are called fractional-order systems (FOSs). These systems are usually described by a coupled system of difference equations, called fractional differential equations (FDEs).

One of the advantages of the fractional derivatives over the integer order is that they describe the holistic evolution of the system that they are modeling; hence, they give a more precise model of the system. For example, fractional calculus intervention can be seen in recent works in the following fields, to name a few: in physics, modelling astrophysics [3],

energy systems [4], and quantum systems [5]; in biology, modeling in cell biology [6], cancer biology [7], and computational biology [8,9]; in engineering, dynamics and control [10,11], fluid mechanics and aerodynamics [12,13], and telecommunication [14,15]. These are some areas where fractional calculus is applied. There are many definitions for fractional derivatives and there exist several numerical and approximate analytical methods for obtaining the approximate solution to the FDEs because most of them do not have an exact solution.

Chaos is a notable phenomenon arising from nonlinear dynamical systems. Basically, chaos is the irregular and unpredictable time evolution of nonlinear dynamical systems whereby trajectories emerging from very close initial points diverge exponentially. The concept of chaos was first discovered, albeit accidentally, by Henri Poincaré, a French mathematician, in the late 1880s while working on the famous *n*-body problem [16]. Several decades later, the chaos concept was officially made popular by Edward Norton Lorenz, a United States mathematician and meteorologist, by his again accidental finding while working on a weather forecast research between 1961 and 1963 [17]. Lorenz's discoveries are twofold. First, the rounding off of the initial condition values to a digit different from the computer's precision generated a considerable difference in the long-term outputs. His atmospheric flow nonlinear model was sensitive to its initial conditions. Second, the three-dimensional (3-D) graphical display of the orbits generated from the model. The plot showed an orderly set of points that are non-repeating, which are now called strange attractors. Consequently, the term "chaos theory", otherwise called the "butterfly effect", was coined in 1975 by James A. Yorke, a mathematician, to describe the chaos phenomenon [18,19].

The study of chaos has revealed the unique properties inherent in the phenomenon [20–23]. Some of these properties include: (1) the sensitivity of the system to, for example, its initial conditions and parameters, and being (2) aperiodic, (3) deterministic, (4) ergodic, and (5) topologically transitive. A chaotic system's qualitative behavior after a large amount of iterations could be an attractor that is either chaotic, periodic, or quasi-periodic depending on the initial conditions' basin of attraction. For the chaotic attractor, its phase space consists of a dense set of unstable periodic orbits (UPOs). Methods such as Pyragas and OGY have been applied to stabilize chaotic nonlinear dynamical systems [24–27]. In addition, the investigations in [28–30] applied feedback control methods to stabilize chaos. Though these methods have been proven to be successful, they have a shortcoming in that they force the UPOs into stable periodic orbits, thereby distorting the original chaotic attractor.

The properties of chaos—for example, the ergodicity and sensitivity of the system to its initial conditions and parameters—are being applied to create different types of systems, most importantly cryptosystems and communication systems. The intervention of chaos in a cryptosystem is significant in this digital world because of the growing need in various sectors to protect confidential and personal information from unauthorized access while they are in storage or in transit over a network. In particular, the proliferation in the usage of images in many sectors, such as military, healthcare, and security, calls for the adequate protection of these sensitive and private images. Consequently, the problem of protecting image information has become an attractive topic for researchers in recent times, giving rise to the proposal of different image encryption methods. In the last few years, many investigations have been performed on chaos-based image encryption and secure communication systems. Examples can be found in [31–39], which are all implemented on the field programmable gate array (FPGA) cards. For electronic implementations, the FPGA has become a better choice over operational amplifiers (OPAMPs) because it is reusable, very cost-effective, faster in performing parallel processing, and faster and efficient with regard to system design. Some other chaos-based image encryption systems implemented numerically and on other devices can be found in [40–47].

In this paper, we present a technique called parameter switching to stabilize fractional-order chaotic spherical systems (FOCSSs). The parameter-switching scheme preserves the underlying chaotic attractor while controlling chaos in the system. Furthermore, we present

an image encryption system, whereby the parameter-switching scheme is applied in the decryption of encrypted RGB and grayscale images. The motivation behind this work is the need for effective methods to stabilize fractional-order systems and to increase the number of works in the literature that apply fractional-order systems in secure communication systems, which, to the best of the authors' knowledge, are few at the moment. This present investigation makes the contributions stated below to improve the state of the art:

(i)   A digital realization of a parameter-switching scheme to stabilize chaos in fractional-order and spherical chaotic nonlinear systems using Grünwald–Letnikov for the numerical approximation and VHDL as the hardware description language.
(ii)  A master–slave-based synchronization of fractional-order and spherical nonlinear systems using the Hamiltonian system with an observer-based approach for the purpose of communication.
(iii) A methodology developed to implement a secure image transmission system on Xilinx and Intel FPGA cards using the parameter-switching technique as a decryption mechanism. The implementation was achieved on two FPGA cards, namely Xilinx's Artix-7 AC701 and Intel's Cyclone V.

The presentation of this work is structured into six sections. Following the introduction is Section 2, which contains the theoretical framework describing the numerical method, models, and techniques relevant to this investigation. Section 3 shows the numerical implementation of the chaos control using the parameter-switching technique for fractional-order chaotic systems and their synchronization using the Hamiltonian system. In Section 4, the digital implementation of chaos control and synchronization using VHDL is also presented. Section 5 contains the implementation on the FPGA of the secure system of image transmission, while we present the conclusion of this investigation in Section 6.

## 2. Theoretical Framework

This section begins by describing the Grünwald–Letnikov numerical method for evaluating fractional-order systems. In addition, the fractional-order chaotic model used in this work is introduced. Furthermore, this section introduces the parameter-switching technique for controlling chaos and also presents the synchronization strategy for the fractional-order chaotic systems.

### 2.1. Grünwald–Letnikov Numerical Method

In fractional-order differential equations, unknown functions are embedded under the operation of fractional-order derivatives. The FDEs generalize differential equations by the application of fractional calculus. Fractional calculus is the generalization of the integer-order differentiation and integration to arbitrary real or complex orders, which are called fractional derivatives and fractional integrals. Given

$$Df(x) = \frac{d}{dx}f(x), \tag{1}$$

where $D$ is the differentiation operator, and

$$If(x) = \int_0^x f(y)dy, \tag{2}$$

where $I$ is the integration operator, in the context of real or complex number powers for Equations (1) and (2), the fractional derivative and fractional integral of the function $f$ are combined into a general, non-integer, and continuous differintegral operator $D^q$, which is defined as:

$$_aD_t^q = \begin{cases} \frac{d^q}{dt^q}, & : q > 0, \\ 1, & : q = 0, \\ \int_a^t (d\tau)^{-q}, & : q < 0, \end{cases} \tag{3}$$

where $q \in \mathbb{R}$, $a$ and $t$ are the bounds of the operator, and $t > a$. It is seen that $D^q$ is a fractional derivative when $q > 0$, fractional integral when $q < 0$, and equal to 1 when $q = 0$.

There are many definitions for the fractional differintegral operator (3). Some of the popular ones are Grünwald–Letnikov [48], Riemann–Liouville [49], and Caputo [50]. In this work, the Grünwald–Letnikov definition and numerical approximation method was adopted to define and provide a solution to the fractional-order chaotic systems. The choice of the Grünwald–Letnikov definition for solving the system of FDEs in this work stems from its convenience with regard to its application. Though the basis is the standard differential operator, it is applicable to the arbitrary order $q$ and uses a discrete addition and binomial coefficient term. Given the forward difference derivative

$$f'(x) = \lim_{x \to 0} \frac{f(x+h) - f(x)}{h}, \tag{4}$$

where $h$ is the step size, higher-order derivatives can be determined recursively. For the second-order derivative, we have:

$$f''(x) = \lim_{x \to 0} \frac{f(x+2h) - 2f(x+h) + f(x)}{h^2}. \tag{5}$$

Using the binomial coefficient $\binom{n}{k}$, where $k \in N$ refers to a term in the polynomial expansion, the formula can be generalized for the $n$-th derivative as follows:

$$f^n(x) = \lim_{h \to 0} \frac{-1^n}{h^n} \sum_{k=0}^{n} (-1)^k \binom{n}{k} f(x + kh). \tag{6}$$

Therefore, if $p \in \mathbb{R}$ replaces the integer $n$ in (6) and the substitution $h$ by $-h$ is made, then the Grünwald–Letnikov direct definition for fractional derivatives is given in the following definition [51].

**Definition 1.** *Let $q \in \mathbb{R}$, where $|q| \leq n$, and let $n = (x-a)/h$; then, the Grünwald–Letnikov fractional derivative is defined as*

$$aD_t^q f(x) = \lim_{h \to 0} \frac{1}{h^q} \sum_{k=0}^{n} (-1)^k \binom{q}{k} f(x - kh). \tag{7}$$

*In the Grünwald–Letnikov definition in (7), the binomial coefficients are $(-1)^k \binom{q}{k}$ for $k = 0, 1, 2, \ldots$, which are calculated as follows:*

$$c_0^{(q)} = 1, \quad c_k^{(q)} = \left(1 - \frac{1+q}{k}\right) c_{k-1}^{(q)}. \tag{8}$$

*When the binomial coefficient is expanded, and using the Gamma function to represent the factorial elements, we have*

$$\binom{q}{k} = \frac{q!}{k!(n-k)!} = \frac{\Gamma(q+1)}{k!\Gamma(q+1-k)}. \tag{9}$$

*Hence, by replacing the binomial coefficient in the definition in (7), the Grünwald–Letnikov derivative becomes:*

$$aD_t^q f(x) = \lim_{h \to 0} \frac{1}{h^q} \sum_{k=0}^{\frac{x-a}{h}} \frac{(-1)^k \Gamma(q+1)}{k!\Gamma(q+1-k)} f(x - kh), \quad \forall\, q \in \mathbb{R}, \quad q \neq -\mathbb{N}_1. \tag{10}$$

*For the numerical evaluation of the q-th derivative of a fractional-order system at the points $tj = jh$, for $j = 1, 2, \ldots$, using the Grünwald–Letnikov definition, the following relation is applied:*

$$aD_{t_j}^q f(x) \approx h^{-q} \sum_{k=0}^{j} \binom{q}{k} f(t_{j-k}). \tag{11}$$

*Therefore, the general numerical approximation using the Grünwald–Letnikov derivative for fractional-order differential equation*

$$aD_*^q x(t) = f(x(t), t) \tag{12}$$

*is written as:*

$$x(t_j) = f(x(t_j), t_j)h^q - \sum_{k=1}^{j} c_k^{(q)} x(t_{j-k}). \tag{13}$$

*Because of the summation operator in the Grünwald–Letnikov derivative, the numerical approximation considers the entire past solutions of the system, from the present point down to the starting point.*

### 2.2. Fractional-Order Chaotic Spherical System

The primary tool in this investigation was the fractional-order model of a 3-D chaotic system proposed in [52], which belongs to a group of dynamical systems that generate spherical attractors. The coupled fractional differential equations of the system are:

$$\left.\begin{aligned}
D_*^{q_1} x &= a_1 x - a_2 y + a_3 z + 2\left(\frac{1 - e^{-200 \sin y}}{1 + e^{-200 \sin y}}\right) \\
D_*^{q_2} y &= -dxz + b + ex \\
D_*^{q_3} z &= c_1 xy + c_2 yz + c_3 z + c
\end{aligned}\right\} \tag{14}$$

where $q_1$, $q_2$, and $q_3$ are the fractional derivative orders, $a_i \neq 0$, $c_i \neq 0$ $(1 \leq i \leq 3)$, $b \neq 0$, $c \neq 0$, $d \neq 0$, and $e$ are all real parameters, and the system-dependent variables are $x$, $y$, and $z$. To numerically evaluate the FOCSS, the Grünwald–Letnikov derivative is applied following the general form given in (13). Therefore, the model in (14) using Grünwald–Letnikov discretization according to Equation (13) becomes:

$$\left.\begin{aligned}
x(t_j) &= \left(a_1 x(t_{i-1}) - a_2 y(t_{i-1}) + a_3 z(t_{i-1}) + 2\left(\frac{1 - e^{-200 \sin y(t_{i-1})}}{1 + e^{-200 \sin y(t_{i-1})}}\right)\right)h^{q_1} - \sum_{k=1}^{j} c_k^{(q_1)} x(t_{j-k}), \\
y(t_j) &= (-dx(t_j)z(t_{j-1}) + b + ex(t_j))h^{q_2} - \sum_{k=1}^{j} c_k^{(q_2)} y(t_{j-k}), \\
z(t_j) &= (c_1 x(t_j)y(t_j) + c_2 y(t_j)z(t_{j-1}) + c_3 z(t_{j-1}) + c)h^{q_3} - \sum_{k=1}^{j} c_k^{(q_3)} z(t_{j-k}),
\end{aligned}\right\} \tag{15}$$

The system parameter values for the above model are $a_1 = -4.1$, $a_2 = 1.2$, $a_3 = 13.45$, $b = 0.161$, and $c = 3.5031$. Others are $c_1 = 2.76$, $c_2 = 0.6$, $c_3 = 13.13$, $d = 1.6$, and $e = 0$. The initial condition is $(x_0, y_0, z_0) = (-0.04, -15.8, -1.4)$. This work considered the incommensurate fractional order; therefore, it is important to find appropriate fractional orders that can make the FOCSS (14) exhibit chaotic behavior. To achieve this goal, a script was created to perform simulation tests on the system with $q_1$, $q_2$, and $q_3$ values between 0 and 1 for step-size $h = 0.01$ and $h = 0.001$. At the end of the tests, $q_1 = 0.9993$, $q_2 = 0.9996$, and $q_3 = 0.9999$ at $h = 0.01$ were selected for this work.

The dynamical behavior of the FOCSS was numerically analyzed using a phase diagram, bifurcation diagram, and Lyapunov exponent (LE) spectra. The LEs $L_1, L_2, \ldots, L_n$,

where $n$ is the dimension of the phase space, were numerically computed using the following formula:

$$L_m \approx \frac{1}{T} \sum_{i=1}^{K} ln||\omega_m^i||,$$

(16)

where $m = 1, 2, \ldots, n$ and $n$ is the dimension of the phase space, $K$ is the orthonormalization time, and $\omega$ is a set of orthonormal vectors obtained by the Gram–Schmidt orthonormalization procedure. For the FOCSS, the LEs are $L_1 = 0.0973$, $L_2 = 0$, and $L_3 = -0.1144$. The bifurcation plot and the Lyapunov exponent spectra are presented in Figure 1.

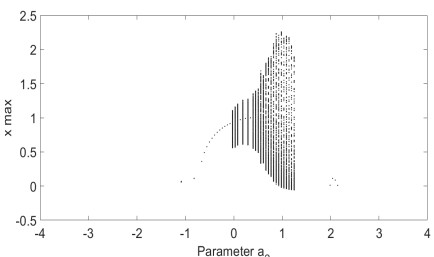 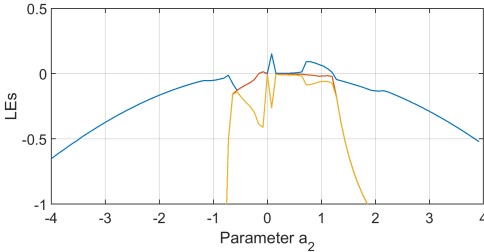

**Figure 1.** Bifurcation diagram and Lyapunov exponent spectra of FOCSS (14), showing the local maxima of state $x$ while varying parameter $a_2$. The LE axis is adjusted to make the three spectral lines more visible.

It is noted that the chaotic attractor was characterized by local instability (chaotic dynamics) and, as the orbits evolved, they formed a closed spherical shape over a long-time simulation. The bifurcation plot and Lyapunov exponent spectra in Figure 1 show the dynamism of the FOCSS (14) and they are in complete agreement. Moreover, it is observed that there is a bifurcation point very close to $a_2 = -0.75$.

### 2.3. Parameter-Switching Technique

Generally, parameter switching is a technique applied to numerically approximate any attractor of an autonomous and continuous nonlinear dynamical system, whether stable or chaotic [31,53]. The technique is based on the theory of Parrondo's paradox applied in game theory, which states that a combination of losing strategies becomes a winning strategy, and vice versa [54]. In the context of the parameter-switching technique and application in chaos theory, chaos represents losing whereas order (stability) represents winning. Hence, we can write:

$$chaos_1 + chaos_2 + \cdots + chaos_N = order,$$

(17)

$$order_1 + order_2 + \cdots + order_N = chaos.$$

(18)

To describe the parameter-switching technique, we start with the following general initial value problem (IVP), describing a large class of fractional-order dynamical systems:

$$D_*^q x(t) = f(x(t)) + pAx(t), x(0) = x_0, \quad \forall \, t \in I = [0, T],$$

(19)

in which $D_*^q$ represents Caputo's deferential operator of order $q$, where $0 < q < 1$, and the continuous nonlinear function is $f : \mathbb{R}^n \to \mathbb{R}^n$. In addition, the control parameter is $p \in \mathbb{R}$, whose values will be switched, the initial condition is $x_0 \in \mathbb{R}^n$, $A$ is a constant matrix, where $A \in L(\mathbb{R}^n)$, and $T > 0$. When $q = 1$ in the IVP (19), the system becomes a classical integer-order IVP, which can be easily evaluated by standard numerical methods

with a fixed step size, such as fourth-order Runge–Kutta. With respect to the FOCSS (14), if parameter $c_3$ is replaced with $p$, we have the following:

$$f(x) = \begin{pmatrix} a_1 x - a_2 y + a_3 z + 2\left(\frac{1-e^{-200\sin y}}{1+e^{-200\sin y}}\right) \\ -dxz + b + ex \\ c_1 xy + c_2 yz + c_3 z + c \end{pmatrix} \tag{20}$$

$$A = \begin{pmatrix} 0 & 0 & 0 \\ 0 & 0 & 0 \\ 0 & 0 & 1 \end{pmatrix} \tag{21}$$

The control parameters $p = p_1, p_2, \cdots, p_N$, with $N > 1$, are deterministically switched over a short time period (integration steps) during which the IVP is integrated numerically, i.e., when $p = p_1$, the IVP is integrated for $w_1$ steps, at $p = p_2$, the IVP is integrated for $w_2$ steps, and so on, up to $p = p_N$ for $w_N$ steps. The switching will keep repeating throughout the entire integration time $I$. Finally, the solution obtained from the IVP by the parameter-switching algorithm, called the "switched" solution and denoted by $A^S$, converges to the "averaged" solution $A^*$ obtained at $p = p^*$ in (19), where $p^*$ represents the average of parameters that are deterministically switched. Parameters $p_i$ and the associated weights $w_i$ must be selected such that the following equation is satisfied:

$$p^* = \frac{\sum\limits_{i=1}^{N} w_i p_i}{\sum\limits_{i=1}^{N} w_i} \tag{22}$$

In the case of obtaining stable cycles (chaos control), the switched parameters $p_i$ and "averaged" parameter $p^*$ are selected from the chaotic regions and the periodic region of the dynamical system, respectively, and there exists at least one periodic region between the chaotic regions. On the other hand, the reverse is the case for obtaining chaotic attractors (chaos anti-control), i.e., the switched parameters are chosen from the periodic regions whereas $p^*$ belongs to a chaotic region, and there is at least one chaotic region between the periodic regions. To computationally verify the results of the parameter-switching scheme, phase diagrams and time series were applied to compare the attractor $A^S$ of the parameter switching with the attractor $A^*$ of the "averaged" solution. For the purpose of this work, the parameter-switching scheme was used to stabilize the chaotic attractors of the FOCSS (14). The approximation of attractors of dynamical systems using the parameter-switching algorithm and its convergence are encapsulated in the following methodology: (1) Replace a system parameter in the IVP with control parameter $p$, which represents the switched parameters; (2) Select the appropriate switched parameters $p_1, p_2, \cdots, p_N$, and their associated weights $w_1, w_2, \cdots, w_N$; (3) Execute the parameter-switching algorithm to numerically evaluate the IVP and generate the "switched" solution $A^S$; (4) Replace the switched parameters $p$ in the IVP with their average value $p^*$, and numerically evaluate the IVP to generate attractor $A^*$ of the "averaged" solution; (5) Verify that $A^S$ matches $A^*$.

**Remark 1.** *The possibility of having different switched parameters p means that different numerical solutions can be obtained.*

**Remark 2.** *With different values for the switched parameters and associated weights, it is possible to obtain the same attractor $A^S$.*

**Remark 3.** *In order to reduce the transient steps, the initial conditions for the "switched" solution and "averaged" solution are the same, without a loss of generality.*

### 2.4. Hamiltonian System

In this section, a synchronization method known as the observer-based Hamiltonian system for synchronizing two fractional-order chaotic systems is presented. In this context, the synchronized systems are in a master–slave relationship, with the master system's states being observed by the slave system. A detailed mathematical description of the method used to model the master and slave FOCSS (14) is as follows: Let

$$D_*^q = f(x) \tag{23}$$

be a fractional-order dynamical system, where the state variable is $D_*^q x \in \mathbb{R}^n$ and the nonlinear function is $f : \mathbb{R}^n \to \mathbb{R}^n$. System (23) can also be written as:

$$D_*^q = A\frac{\partial H}{\partial x} + \mathcal{F}(x), \tag{24}$$

in which $A = \frac{A-A^T}{2} + \frac{A+A^T}{2}$ and $\mathcal{F}(x)$ is a destabilizing vector field, whereby $A$ and $\mathcal{F}(x)$ represent the fractional-order system equations. Hence,

$$D_*^q = \frac{A - A^T}{2}\frac{\partial H}{\partial x} + \frac{A + A^T}{2}\frac{\partial H}{\partial x} + \mathcal{F}(x). \tag{25}$$

If $\mathcal{J}(x) = \frac{A-A^T}{2}$ and $\mathcal{S}(x) = \frac{A+A^T}{2}$, Equation (24) can then be transformed into a generalized Hamiltonian canonical system as follows:

$$D_*^q = \mathcal{J}\frac{\partial H}{\partial x} + \mathcal{S}\frac{\partial H}{\partial x} + \mathcal{F}(x), \quad x \in \mathbb{R}^n, \tag{26}$$

where $H(x) = \frac{1}{2}x^T \mathcal{M}x$ is a positive smooth energy function definite in $\mathbb{R}^n$ and $\mathcal{M}$ is a constant, symmetric, positive definite matrix. Therefore, $\frac{\partial H}{\partial x} = \mathcal{M}x$, which is the column gradient vector of $H(x)$. Matrix $\mathcal{J}(x)$ satisfies $\mathcal{J}(x) + \mathcal{J}^T(x) = 0$ and $\mathcal{S}(x)$ satisfies $\mathcal{S}(x) = \mathcal{S}^T(x)$ for all $x \in \mathbb{R}^n$. The vector field, denoted by $\mathcal{J}(x)\frac{\partial H}{\partial x}$, exhibits the conservative aspect of the system, whereas $\mathcal{S}(x)$ shows the non-conservative part.

Now, the master system, or simply the master, is created with a destabilizing vector field and an output $y(t)$ as shown in Equation (27):

$$\left.\begin{aligned} D_*^q x &= \mathcal{J}(y)\frac{\partial H}{\partial x} + \mathcal{S}(y)\frac{\partial H}{\partial x} + \mathcal{F}(y), \quad x \in \mathbb{R}^n, \\ y &= \mathcal{C}\frac{\partial H}{\partial x}, \quad y \in \mathbb{R}^m, \end{aligned}\right\} \tag{27}$$

where $\mathcal{S}$ is a constant symmetric matrix and $\mathcal{C}$ is a constant matrix.

For the slave system, or simply the slave, the estimate of state vector $x$ is represented by $\xi$, whereas the estimated output is represented by $\eta$. Hence, the slave system is denoted by:

$$\left.\begin{aligned} D_*^q \xi &= \mathcal{J}(y)\frac{\partial H}{\partial \xi} + \mathcal{S}(y)\frac{\partial H}{\partial \xi} + \mathcal{F}(y) + g_i\phi_i, \quad \xi \in \mathbb{R}^n, \\ \eta &= \mathcal{C}\frac{\partial H}{\partial \xi}, \quad \eta \in \mathbb{R}^m, \end{aligned}\right\} \tag{28}$$

where $g_i$ is the observer gain and $\phi_i = y - \eta$ is the output estimation error, with $i = 1, 2, 3$. Hence, $\phi = x - \xi$ is the state estimation error.

Generally, the status of the synchronization of the two systems, successful or otherwise, is determined by the conditions given in Definition 1 and Theorems 3 and 4 in [32]. Based on the Hamiltonian synchronization system analysis, the master and slave systems of the FOCSS (14) is constructed as follows:

$$\mathcal{J}(x) = \begin{bmatrix} 0 & \frac{-a_2-e}{2} & \frac{a_3}{2} \\ \frac{a_2+e}{2} & 0 & \frac{-c_1x-dx}{2} \\ -\frac{a_3}{2} & \frac{c_1x-dx}{2} & 0 \end{bmatrix} \tag{29}$$

$$\mathcal{S}(x) = \begin{bmatrix} a_1 & \frac{-a_2+e}{2} & \frac{a_3}{2} \\ \frac{-a_2+e}{2} & 0 & \frac{c_1x-dx}{2} \\ \frac{a_3}{2} & \frac{c_1x-dx}{2} & c_2y+c_3 \end{bmatrix} \tag{30}$$

and

$$\mathcal{F}(x) = \begin{bmatrix} 2\left(\frac{1-e^{-200\sin y}}{1+e^{-200\sin y}}\right) \\ b \\ c \end{bmatrix} \tag{31}$$

From Equation (27),

$$\begin{bmatrix} D_*^{q_1}x \\ D_*^{q_2}y \\ D_*^{q_3}z \end{bmatrix} = \begin{bmatrix} 0 & \frac{-a_2-e}{2} & \frac{a_3}{2} \\ \frac{a_2+e}{2} & 0 & \frac{-c_1x-dx}{2} \\ -\frac{a_3}{2} & \frac{c_1x-dx}{2} & 0 \end{bmatrix}\frac{\partial H}{\partial x} + \begin{bmatrix} a_1 & \frac{-a_2+e}{2} & \frac{a_3}{2} \\ \frac{-a_2+e}{2} & 0 & \frac{c_1x-dx}{2} \\ \frac{a_3}{2} & \frac{c_1x-dx}{2} & c_2y+c_3 \end{bmatrix}\frac{\partial H}{\partial x} + \begin{bmatrix} 2\left(\frac{1-e^{-200\sin y}}{1+e^{-200\sin y}}\right) \\ b \\ c \end{bmatrix} \tag{32}$$

where $H(x) = \frac{1}{2}[x^2 + y^2 + z^2]$, with the gradient vector $\frac{\partial H}{\partial x} = \begin{bmatrix} x \\ y \\ z \end{bmatrix}$.

Hence, the master system is written as follows:

$$\begin{bmatrix} D_*^{q_1}x \\ D_*^{q_2}y \\ D_*^{q_3}z \end{bmatrix} = \begin{bmatrix} a_1x - a_2y + a_3z \\ -dxz + ex \\ -c_1xy + c_2yz + c_3z \end{bmatrix} + \begin{bmatrix} 2\left(\frac{1-e^{-200\sin y}}{1+e^{-200\sin y}}\right) \\ b \\ c \end{bmatrix} \tag{33}$$

Furthermore, from Equation (28),

$$\begin{bmatrix} D_*^{q_1}u \\ D_*^{q_2}v \\ D_*^{q_3}r \end{bmatrix} = \begin{bmatrix} 0 & \frac{-a_2-e}{2} & \frac{a_3}{2} \\ \frac{a_2+e}{2} & 0 & \frac{-c_1u-du}{2} \\ -\frac{a_3}{2} & \frac{c_1u-du}{2} & 0 \end{bmatrix}\frac{\partial H}{\partial \xi} + \begin{bmatrix} a_1 & \frac{-a_2+e}{2} & \frac{a_3}{2} \\ \frac{-a_2+e}{2} & 0 & \frac{c_1u-du}{2} \\ \frac{a_3}{2} & \frac{c_1u-du}{2} & c_2v+c_3 \end{bmatrix}\frac{\partial H}{\partial \xi} + \begin{bmatrix} 2\left(\frac{1-e^{-200\sin v}}{1+e^{-200\sin v}}\right) \\ b \\ c \end{bmatrix} + \begin{bmatrix} g_1 \\ g_2 \\ g_3 \end{bmatrix}\phi_i \tag{34}$$

and the slave system is simplified to become

$$\begin{bmatrix} D_*^{q_1}u \\ D_*^{q_2}v \\ D_*^{q_3}r \end{bmatrix} = \begin{bmatrix} a_1u - a_2v + a_3r \\ -dur + eu \\ -c_1uv + c_2vr + c_3r \end{bmatrix} + \begin{bmatrix} 2\left(\frac{1-e^{-200\sin v}}{1+e^{-200\sin v}}\right) \\ b \\ c \end{bmatrix} + \begin{bmatrix} g_1 \\ g_2 \\ g_3 \end{bmatrix}\phi_i \tag{35}$$

where $g_1$, $g_2$, and $g_3$ are the observer gains, and $\phi_i$ is the error arising from the synchronization.

It is seen that the master is the original FOCSS (14), but the slave system is modeled to receive feedbacks in each iteration with the error already being compensated. Both systems are expressed below in Grünwald–Letnikov derivative form:

$$x(t_j) = \left( a_1 x(t_{i-1}) - a_2 y(t_{i-1}) + a_3 z(t_{i-1}) + 2\left( \frac{1 - e^{-200 \sin y(t_{i-1})}}{1 + e^{-200 \sin y(t_{i-1})}} \right) \right) h^{q_1} - \sum_{k=1}^{j} c_k^{(q_1)} x(t_{j-k}),$$

$$y(t_j) = \left( -dx(t_j)z(t_{j-1}) + b + ex(t_j) \right) h^{q_2} - \sum_{k=1}^{j} c_k^{(q_2)} y(t_{j-k}),$$

$$z(t_j) = \left( c_1 x(t_j)y(t_j) + c_2 y(t_j)z(t_{j-1}) + c_3 z(t_{j-1}) + c \right) h^{q_3} - \sum_{k=1}^{j} c_k^{(q_3)} z(t_{j-k}),$$

(36)

$$u(t_j) = \left( a_1 u(t_{i-1}) - a_2 v(t_{i-1}) + a_3 r(t_{i-1}) + 2\left( \frac{1 - e^{-200 \sin v(t_{i-1})}}{1 + e^{-200 \sin v(t_{i-1})}} \right) + g_1\phi_1 \right) h^{q_1} - \sum_{k=1}^{j} c_k^{(q_1)} u(t_{j-k}),$$

$$v(t_j) = \left( -du(t_j)r(t_{j-1}) + b + eu(t_j) + g_2\phi_2 \right) h^{q_2} - \sum_{k=1}^{j} c_k^{(q_2)} v(t_{j-k}),$$

$$r(t_j) = \left( c_1 u(t_j)v(t_j) + c_2 v(t_j)r(t_{j-1}) + c_3 r(t_{j-1}) + c + g_3\phi_3 \right) h^{q_3} - \sum_{k=1}^{j} c_k^{(q_3)} r(t_{j-k}),$$

(37)

## 3. Numerical Implementations

This section presents the results of the numerical implementations of chaos control in the fractional-order chaotic system (14) using the parameter-switching technique and the synchronization of the chaotic system using the Hamiltonian system. The numerical implementations were performed in Matlab version R2020b.

### 3.1. Chaos Control in FOCSS

By following the five-step methodology stated in Section 2.3 for synthesizing desired attractors of dynamical systems with the parameter-switching technique, a stable solution of the FOCSS (14) can be obtained. When a considerably long transient period is neglected, the solution obtained by the parameter switching will approximate a unique stabilized solution in the same basin of attraction.

#### 3.1.1. Case 1: Four Control Parameters

The process begins by replacing parameter $c_3$ in FOCSS (14) with a control parameter $p$ and, in this implementation, the number of control parameters is $N = 4$. Therefore, let

- $P_N = \{p_1, p_2, p_3, p_4\} = \{10.05, 10.18, 13.00, 13.05\}$, which are selected from the chaotic regions of the FOCSS (14) (see the bifurcation diagram in Figure 2);
- $W_N = \{w_1, w_2, w_3, w_4\} = \{2, 1, 2, 1\}$, which are the corresponding weights of $P_N$; and
- $A_N = \{A^{p_1}, A^{p_2}, A^{p_3}, A^{p_4}\}$, which are the chaotic attractors corresponding to $P_N$.

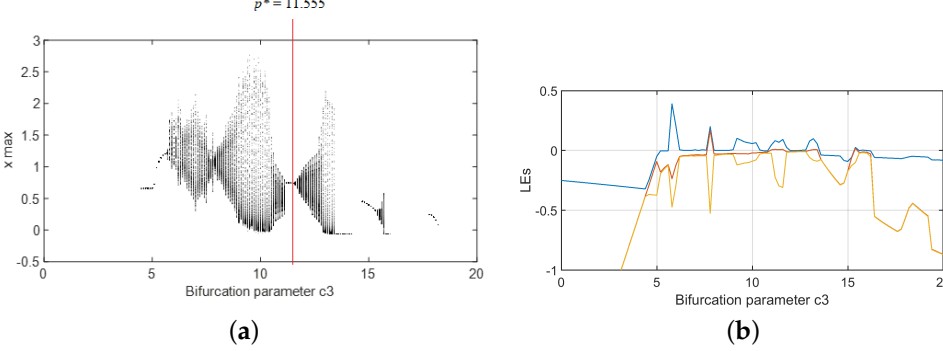

**Figure 2.** (**a**) Bifurcation diagram of FOCSS (14) with local maxima of state $x$, showing the location $p^*$ and (**b**) the corresponding Lyapunov exponent spectra. The LE axis is adjusted to make the three spectral lines more visible.

It is noted that both $P_N$ and $W_N$ were carefully chosen such that Equation (22) is satisfied, whereby $p^* = 11.555$, which corresponds to a stable window of the FOCSS (14)

(see Figure 2) located in the real open interval $(p_1, p_4)$. Three-dimensional phase plots (removing the transient time) and time series were applied to numerically verify that the "averaged" solution $A^*$, obtained when parameter $c_3$ was replaced with $p^*$ in the FOCSS (14), was approximated by the "switched" solution $A^S$. Figure 3 contains the chaotic attractors $A_N$ of $P_N$ and their state $x$ time series. Figure 4 shows the attractors $A^S$ (red) and $A^*$ (blue) and the time series of both $A^S$ and $A^*$.

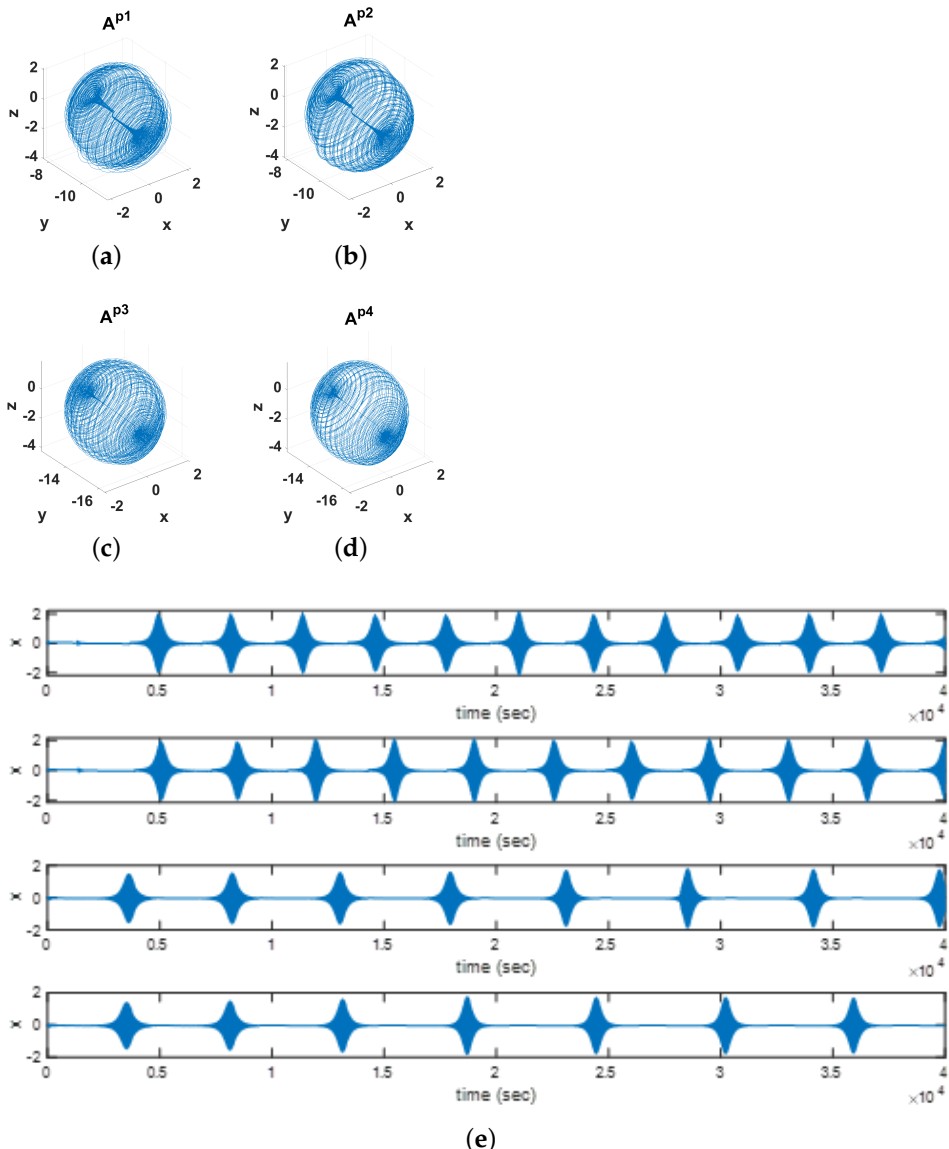

**Figure 3.** (**a**–**d**) Chaotic attractors $A^{p_1}$, $A^{p_2}$, $A^{p_3}$, and $A^{p_4}$ of the FOCSS (14) from parameters $p_1 = 10.05$, $p_2 = 10.18$, $p_3 = 13.00$, and $p_4 = 13.05$, respectively, and (**e**) state $x$ time series of the attractors.

The FOCSS (14) is a nonlinear system; and the effect of the nonlinearity and other factors, such as the sensitivity to its initial conditions, parameters, and fractional orders, is seen in Figure 3, whereby the attractors $A^{p_1}$, $A^{p_2}$, $A^{p_3}$, and $A^{p_4}$ are chaotic. In a way, it can be said that these attractors were produced by four subsystems of (14), i.e., when $c_3 = [p_1, p_2, p_3, p_4]$. However, chaos in the underlying system was controlled, producing a stable cycle shown in Figure 4a, after the four subsystems were repeatedly evaluated by the parameter-switching algorithm. The verification of the effectiveness of the parameter-switching technique shows that the "switched" solution matches the "averaged" solution (Figure 4b). It is seen in the superimposed plot in Figure 4c that the synthesized stable

attractor $A^S$ perfectly matches attractor $A^*$, and in Figure 4d, where the time series of both solutions match each other.

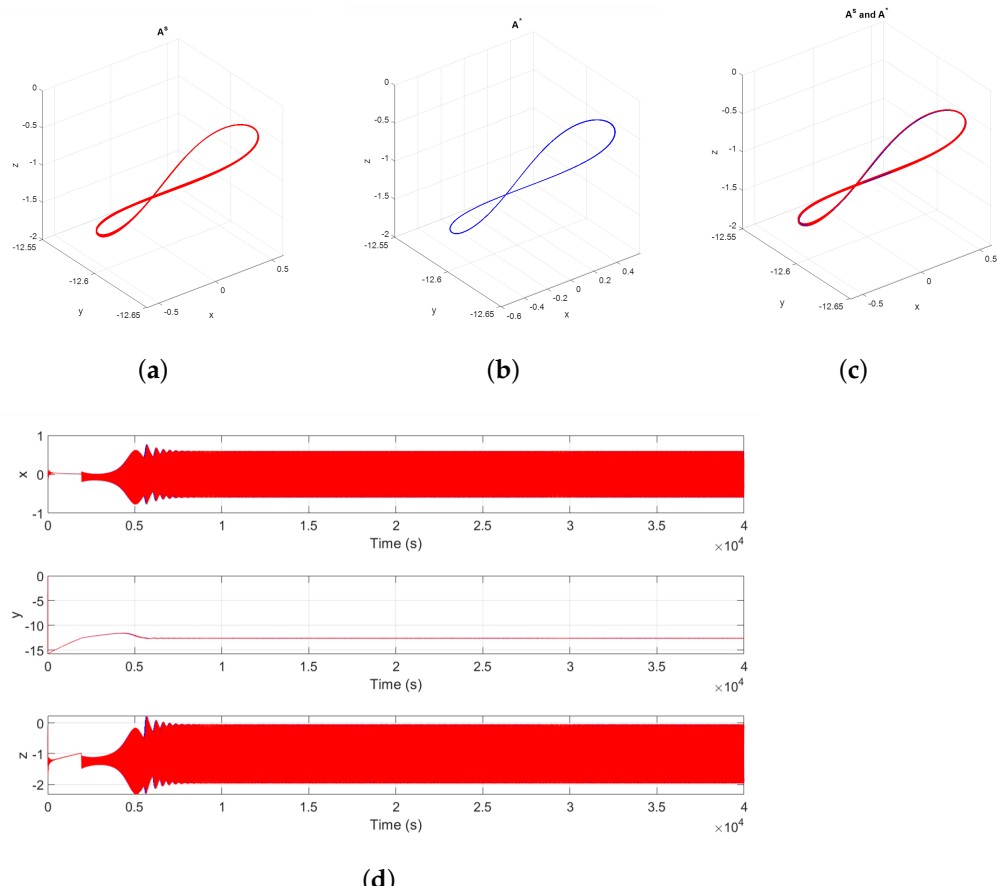

**Figure 4.** Matlab verification of parameter-switching scheme for four control parameters: (**a**) "Switched" solution $A^S$; (**b**) "averaged" solution $A^*$; (**c**) superimposition of $A^S$ on $A^*$; (**d**) superimposition of time series of state variables $x$, $y$, and $z$ of $A^S$ (red) and $A^*$ (blue).

3.1.2. Case 2: Six Control Parameters

In this case, the control parameter $p$ in the FOCSS (14) includes six parameters for the chaos control, whereby $N = 6$. Therefore, let

- $P_N = \{p_1, p_2, p_3, p_4, p_5, p_6\} = \{10.18, 10.34, 10.56, 13.00, 13.05, 13.125\}$, which are selected from the chaotic regions of the FOCSS (14);
- $W_N = \{w_1, w_2, w_3, w_4, w_5, w_6\} = \{2, 1, 2, 2, 1, 1\}$, which are the corresponding weights of $P_N$; and
- $A_N = \{A^{p_1}, A^{p_2}, A^{p_3}, A^{p_4}, A^{p_5}, A^{p_6}\}$, which are the chaotic attractors corresponding to $P_N$.

As usual, both $P_N$ and $W_N$ were carefully chosen such that Equation (22) is satisfied, whereby $p^* = 11.555$, which corresponds to a stable window of the FOCSS (14) located in the real open interval $(p_1, p_6)$. Phase plots, together with time series, were applied to numerically verify that $A^*$, obtained when parameter $c_3$ was replaced with $p^*$ in the FOCSS (14), was approximated by $A^S$ of the parameter-switching algorithm. Figure 5 presents the chaotic attractors $A_N$ of $P_N$ and their time series. Figure 6 shows the attractors $A^S$ (red) and $A^*$ (blue) and the time series of both $A^S$ and $A^*$.

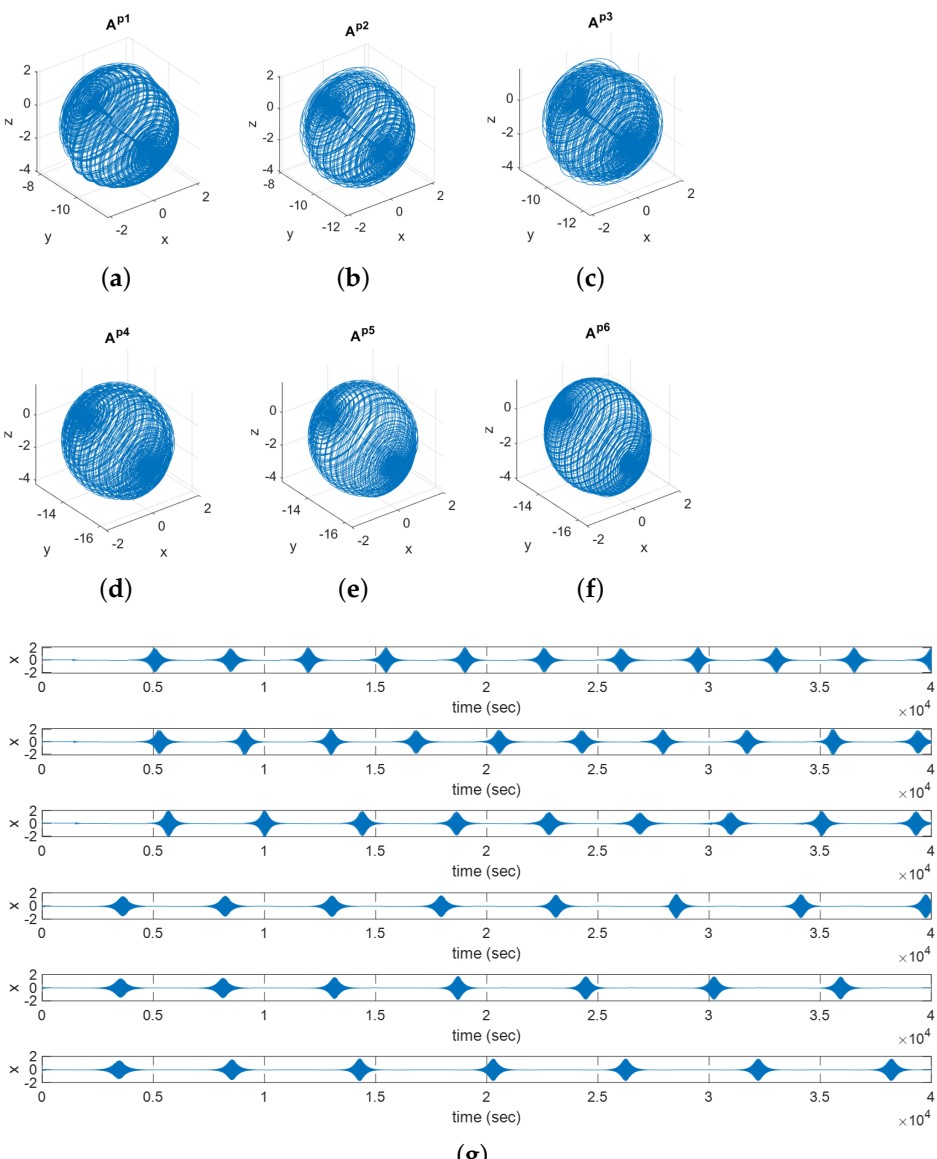

**Figure 5.** (**a**–**f**) Chaotic attractors $A^{p_1}$, $A^{p_2}$, $A^{p_3}$, $A^{p_4}$, $A^{p_5}$, and $A^{p_6}$ of the FOCSS (14) from parameters $p_1 = 10.18$, $p_2 = 10.34$, $p_3 = 10.56$, $p_4 = 13.00$, $p_5 = 13.05$, and $p_6 = 13.125$, respectively, and (**g**) state $x$ time series of the attractors.

Similar to Case 1, chaos in the system was stabilized with the parameter-switching technique, producing the stable cycle in Figure 6a. The parameter-switching solution was verified against the "averaged" solution (Figure 6b) as shown in the matching superimposed attractors in Figure 6c. Further verification is given in the superimposed time series in Figure 6d.

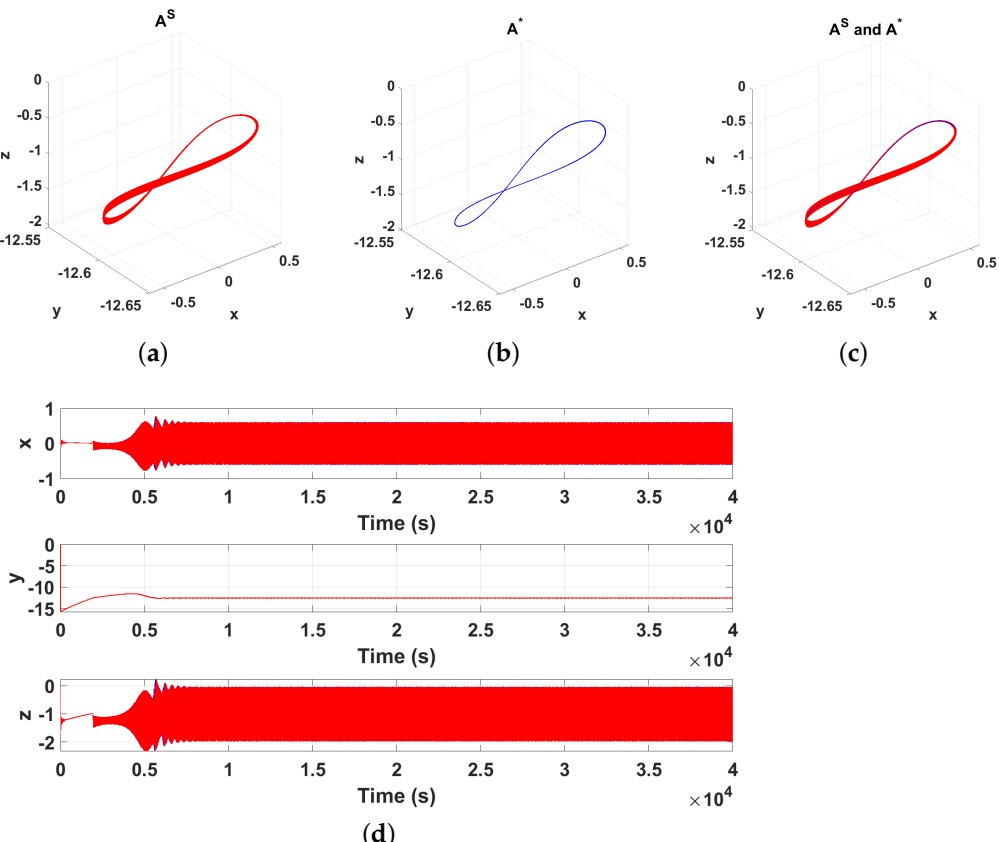

**Figure 6.** Matlab verification of parameter-switching scheme for six control parameters: (**a**) "switched" solution $A^S$; (**b**) "averaged" solution $A^*$; (**c**) superimposition of $A^S$ on $A^*$; (**d**) superimposition of time series of state variables $x$, $y$, and $z$ of $A^S$ (red) and $A^*$ (blue).

### 3.2. Synchronization of FOCSSs

Based on the Hamiltonian synchronization system analysis in Section 2.4, any of the switched parameters could replace parameter $c_3$ in both the master (33) and slave (35). The gains of the slave system were $g_1 = 2$, $g_2 = 7$, and $g_3 = 5$. The initial condition of the master system was $(x_0, y_0, z_0) = (-0.04, -15.8, -1.4)$, whereas, for the slave system, the initial condition was $(u_0, v_0, r_0) = (-0.12, -12.41, -2.1)$. Regardless of what values are assigned to the initial conditions, the synchronization is considered successful when $\lim_{t \to \infty} ||x(t) - \xi(t)|| = 0$. To examine the status of the synchronization, the estimation errors between the master and slave states were computed as thus: $\phi_1 = x - u$, $\phi_2 = y - v$, and $\phi_3 = z - r$. The numerical simulation of both systems are presented in Figure 7a,b. The state estimation errors $\phi_1$, $\phi_2$, and $\phi_3$ are shown in Figure 7c–e and the phase errors are plotted in Figure 7f–h.

It is seen in Figure 7 that the strange attractors of both systems are identical. The synchronization of the two system was successful as seen in the state estimation and phase errors in Figure 7c–e and Figure 7f–h, respectively. In fact, the two systems converged within a very short time after the simulation began, making $(x_i, y_i, z_i) = (u_i, v_i, r_i)$, and the state estimation errors $\phi_1$, $\phi_2$, and $\phi_3$ became zero (see Figure 7c–e). The convergence of the slave to the master can also be verified in the phase error (see Figure 7f–h), where the corresponding states were plotted against each other. What this means is that the two systems behaved alike after they were completely synchronized. These results were obtained for $c_3 = 13.05$ in both systems, which is similar to when parameter $c_3$ takes other values of $P_N$.

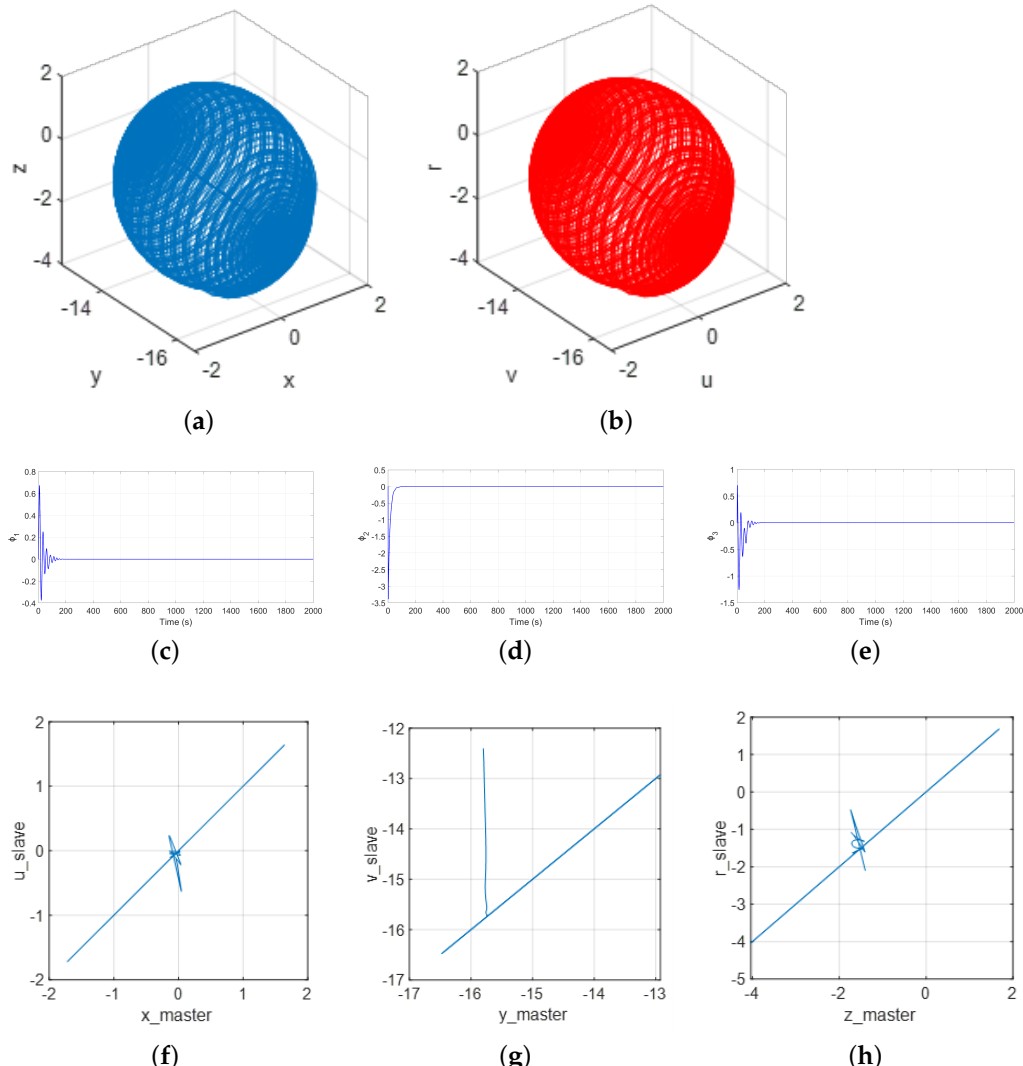

**Figure 7.** Synchronization when $c_3 = 13.05$ in Matlab: (**a**) master system (33); (**b**) slave system (35); (**c**–**e**) state estimation errors $\phi_1 = x - u$, $\phi_2 = y - v$, and $\phi_3 = z - r$ between the master system (33) and slave system (35); (**f**–**h**) phase errors between $x$, $y$, and $z$ of master system (33) and corresponding states $u$, $v$, and $r$ of slave system (35).

## 4. VHDL Implementations

The digital realizations of the chaos control and synchronization in VHDL based on the FOCSS (14) are presented in this section. The hardware design was based on a 32-bit number format, which is composed of 1 bit for the sign, 6 bits for the integer component, and 25 bits for the fractional part. The fixed-point computations were made possible by two IEEE libraries, namely $fixed\_pkg$ and $fixed\_float\_types$, whereas communication of data among entities was performed with the $sfixed$ type. To simplify the hardware implementation of the FOCSS (14) and reduce the computational intensity, we considered avoiding the computation of the hyperbolic tangent function in the model on the FPGA, which consists of the sine and exponential functions $sin(y)$ and $e^y$, respectively. Instead, we chose to create a look-up table (LUT) for the function. This option results in a single LUT of 512 samples of 8 bits each. The VHDL implementations used the same parameter values and initial conditions as the Matlab simulations in the previous section.

### 4.1. Chaos Control in FOCSS

The VHDL design for chaos control in the fractional-order chaotic spherical system using the parameter-switching technique consists of one entity, one architecture, and two processes. Figure 8 depicts the general block representation of the design, showing the entity and the two processes. One of the two processes, called *SWITCHER*, serves as a multiplexer used to control the switching of parameter $p$ between $p_1$, $p_2$, $p_3$, and $p_4$ in Case 1 and $p_1$, $p_2$, $p_3$, $p_4$, $p_5$, and $p_6$ in Case 2 for the FOCSS in accordance with their respective associated weights. On the other hand, the *SOLVER* process performs the approximation of the FOCSS using the Grünwald–Letnikov method to generate the output signals $x_i$, $y_i$, and $z_i$, whereas $x_0$, $y_0$, and $z_0$ are the initial conditions for the FOCSS. The entity *SWITCHING_P* has one input signal *clk* and three output signals $x\_0$, $y\_0$, and $z\_0$ representing the stabilized states of the FOCSS.

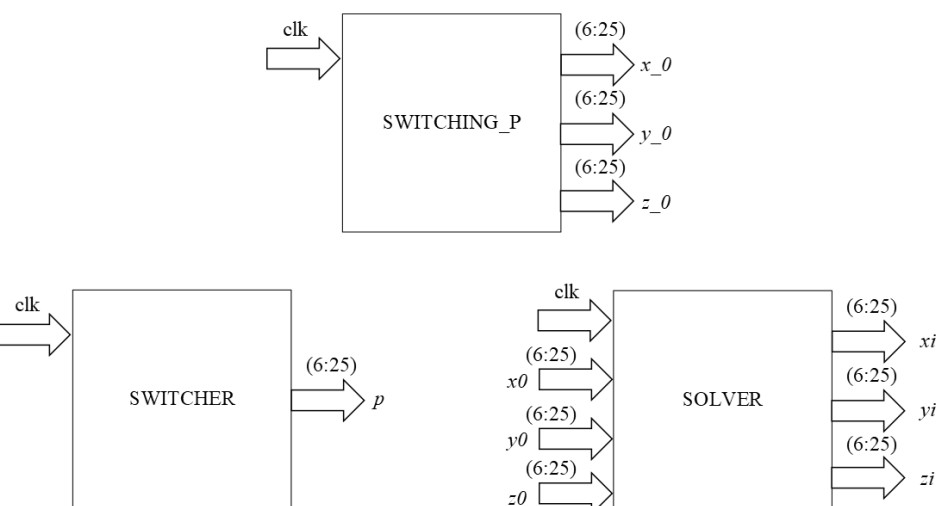

**Figure 8.** Block description of the parameter-switching entity *SWITCHING_P*, process *SWITCHER*, and process *SOLVER*.

The VHDL implementations showed that the "switched" solution produced attractors with periodic orbits after the transient time, meaning that chaos in the system has been stabilized. The Case 1 output of the parameter-switching technique was visualized on the TiePie Handyscope HS3 oscilloscope as shown in Figure 9. The plots obtained for Case 2 on the oscilloscope are similar to Case 1.

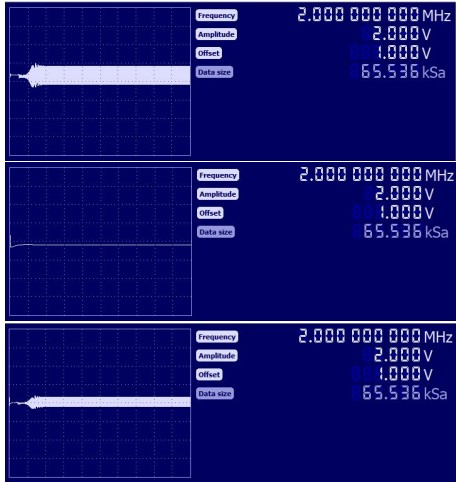

**Figure 9.** Output of the parameter-switching technique Case 1 as visualized on TiePie Handyscope HS3 oscilloscope.

### 4.2. Synchronization of FOCSSs

To implement the digital counterpart of the master and slave synchronization simulated in Matlab in VHDL, four entities were created. The first and second entities describe the master (33) and slave (35) systems, respectively. The third entity describes the computation of error compensation for the slave system using the Hamiltonian synchronization strategy. The fourth, which is the main (top) synchronization entity, connects with the previous three entities. All four entities operate in parallel. The implementation block diagram of the synchronization system is shown in Figure 10. For the co-simulation performed in Active-HDL, the clock period was 0.5 s, while the total synchronization time was 40,000 s.

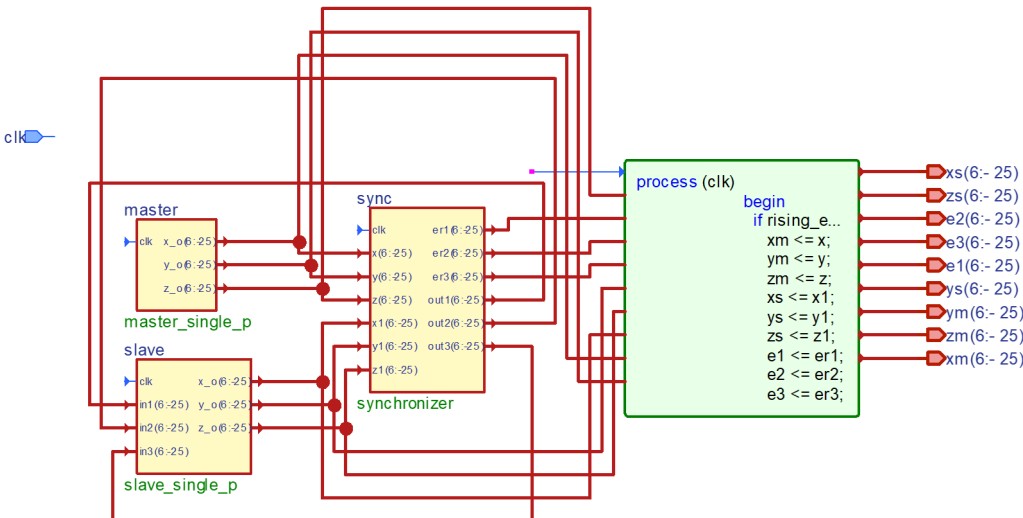

**Figure 10.** Block diagram showing the VHDL implementation of synchronized master system (33) and slave system (35).

In the block diagram above, the entities describing the master system, slave system, and error compensation are represented by *master*, *slave*, and *sync* units, respectively. The *slave* unit receives feedbacks and compensates for the error in each state in every iteration. The output signals of the implementation were obtained from the top entity. The output signals consist of $x_m$, $y_m$, and $z_m$ as chaotic states of the master and $x_s$, $y_s$, and $z_s$ as chaotic states of the slave, and the estimation errors are $e_1$, $e_2$, and $e_3$, calculated in the *sync* unit from the three states of the master and slave. Chaotic states of both the master and slave are displayed in Figure 11a, whereas the state estimation errors are presented in Figure 11b.

It is observed in Figure 11a that both the master and slave systems have identical dynamical behavior. More importantly, by comparing the state estimation errors of the VHDL implementation in Figure 11b with the Matlab simulations in Figure 7c–e, it is noted that both implementations agree as $(x_m, y_m, z_m) = (x_s, y_s, z_s)$. Hence, the errors converge to zero, meaning that the digital implementation was successful. Moreover, the state estimation errors were visualized on the TiePie Handyscope HS3 oscilloscope as shown in Figure 12.

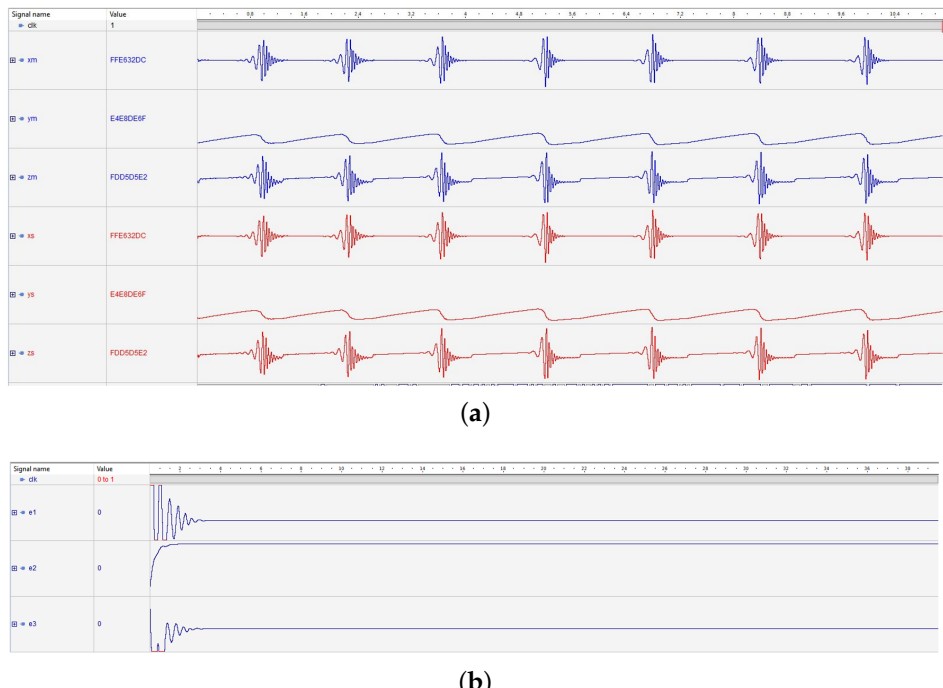

**(a)**

**(b)**

**Figure 11.** VHDL implementation when $c_3$ = 13.05; (**a**) chaotic states of the master (33) in blue color and slave (35) in red color; (**b**) state estimation errors $e_1$, $e_2$, and $e_3$ between the master and slave systems.

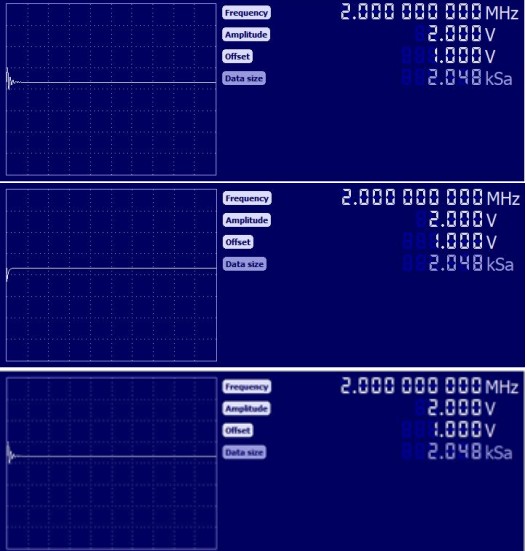

**Figure 12.** State estimation errors $e_1$, $e_2$, and $e_3$ as visualized on TiePie Handyscope HS3 oscilloscope.

## 5. Secure Image Transmission System

Much evidence exists in literature that the secure communication system remains the area where chaos theory has been most applied [31,32,35,55], exploiting the properties of chaos; for example, its ergodicity and sensitivity to the initial conditions and parameters of the system. The pseudo-random sequence of numbers generated by chaotic systems is suitable for hiding and carrying data across a communication channel. In this section, we present a secure system for image transmission, implemented to encrypt and send grayscale and RGB images. The system is a unification of the synchronized master–slave systems, encryption/decryption system, and parameter-switching algorithm.

Once the chaotic systems synchronization is achieved, it is possible to encrypt and transmit confidential data from the master to the slave. The ability of unauthorized persons to decipher the encrypted data would be very limited because chaotic oscillators are very sensitive to the keys, which, in this work, include the parameters, initial conditions, and fractional orders. In the encryption algorithm, the plain image, e.g., RGB or grayscale image, undergoes one round of encryption using any of the three chaotic states of the master system, which is followed by a diffusion process, resulting in ciphered data. However, it is important for the vector size $X$ of the chaotic state to not be less than the vector size $I$ of the image pixels, i.e., size $(X) \geq$ size $(I)$. In this work, a 24-bit depth RGB image of size $320 \times 240$ pixels (230,400 words) and an 8-bit depth grayscale image of size $640 \times 480$ pixels (307,200 words) were used as case studies. The encryption algorithm is described in the flowchart in Figure 13.

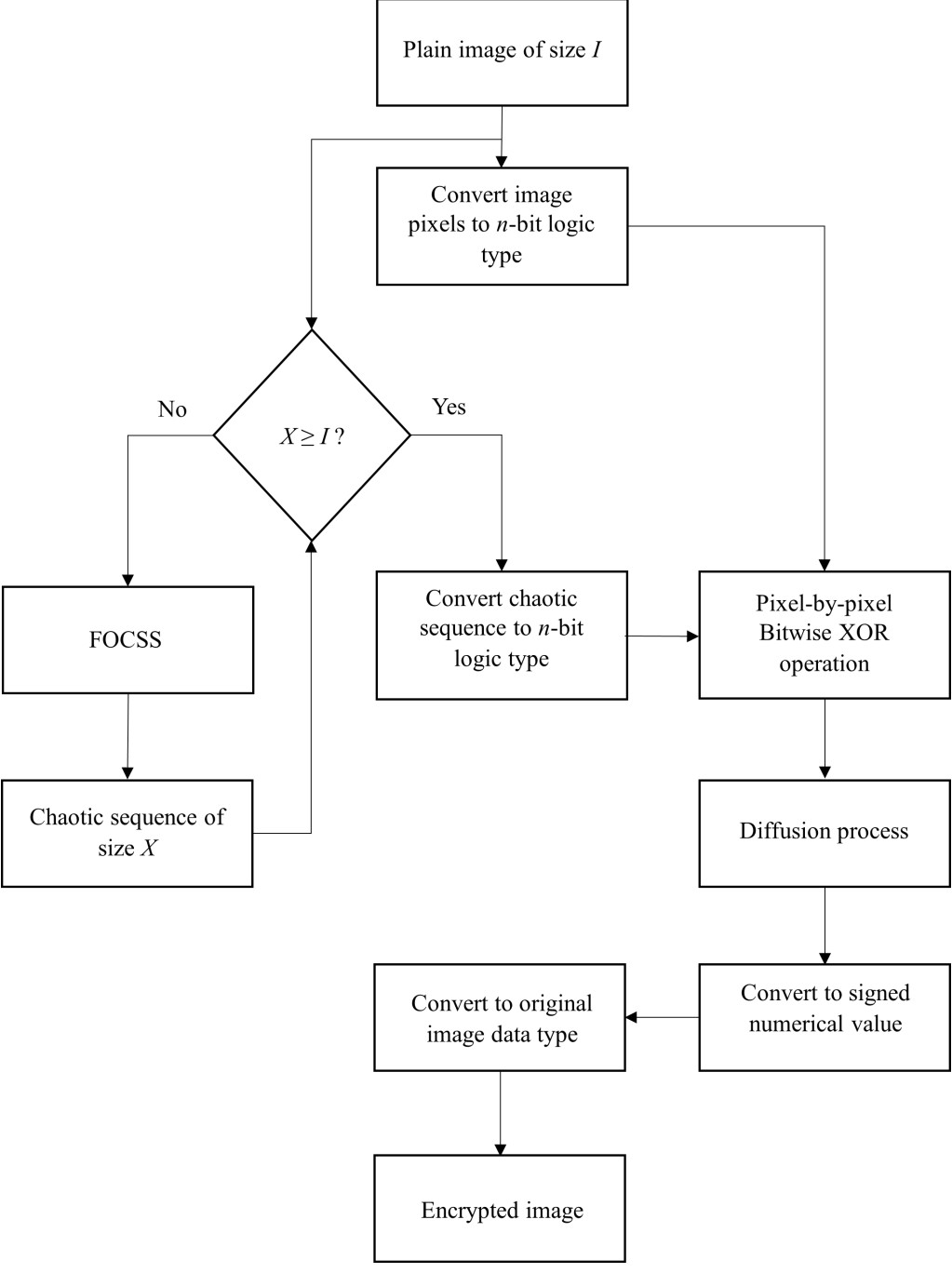

**Figure 13.** Chaos-based encryption algorithm.

In the algorithm, the size of the plain image *I* is compared with the size of the generated chaotic sequence *X* to ensure that the latter is not smaller than the former. The image pixels and elements of the chaotic sequence are each converted to *n*-bit logic data type, which was 9-bit in this work. Thereafter, a bitwise XOR operation is performed between the image pixels and the chaotic sequence, which is followed by the diffusion of the intermediate encrypted data. To be able to compare the plain and the encrypted images, the latter is converted back to the data type of the former.

The digital implementation of the secure system for image transmission is on Artix-7 AC701 using Xilinx with device number XC7A200T-2FBG676C and Intel's Cyclone V with device number 5CGXFC9E7F35C8. It has been established that once the master and slave systems synchronize, the transmission of information is possible. For the encryption and decryption system, the initial conditions, system parameters, and fractional orders are private keys; hence, the system is symmetric.

The implemented image transmission system is simpler in architecture due to its few subsystems. In the block diagram presented in Figure 14, the synchronized system, consisting of the *master*, *slave*, and *sync* units, is the backbone of the system. Technically, the slave unit consists of the slave system (35) with chaotic output signals *x_sync*, *y_sync*, and *z_sync*, and an embedded parameter-switching algorithm with stabilized output signals $x_o$, $y_o$, and $z_o$. The *txn_rxn* unit coordinates the transmission system, which includes the encryption and decryption operations. The outputs are received via the signals *image_original* for the original images, *encrypted* for the encrypted images, and *image_out* for the decrypted images. The transmission system was used to encrypt, send, and decrypt RGB and grayscale images. The master system (33) is the transmitter, where the images are encrypted by any of the chaotic states *x_o*, *y_o*, or *z_o* and sent to the receiver, the slave system, where decryption is performed by the parameter-switching technique.

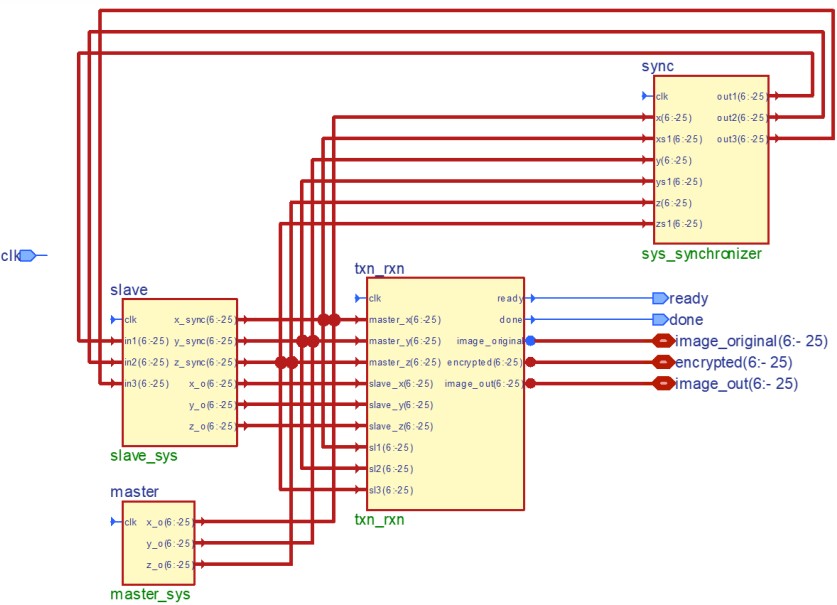

**Figure 14.** Block diagram showing the VHDL implementation of a secure image transmission system.

In this work, the secure transmission of images are summarized in the following methodology: (1) Create and store the original image pixel in VHDL format; (2) Start the synchronization of the transmitter, which is the master, and the receiver, which is the slave; (3) Set flag *ready* to 1 when the transmitter and receiver synchronize; (4) Enter the encryption, sending, and decryption phase when flag *ready* is 1; (5) Encrypt the original image pixel with the chaotic master state using the XOR operation followed by a diffusion process; (6) Pass the encrypted image data into the receiver for decryption; (7) Decrypt the encrypted image data using the master state stabilized by the parameter-switching

technique; (8) Set flag *done* to 1 if all the image pixels have been encrypted and decrypted; (9) Reshape the decrypted image vector into the size of the original image.

The experiment used a universal asynchronous receiver–transmitter (UART) communication module, which was implemented in the *txn_rxn* unit and consists of three states, namely *wait*, *encrypt*, and *download*. The *wait* state is when the system loops continuously until an image input packet is received via the UART module, and, thereafter, the process jumps into the *encrypt* state. At the *encrypt* state, the encryption of the image, which has been resized and converted into 9 bits, is performed with a 9-bit chaotic state using an XOR operation. In addition, the decryption of the image occurs at this state. The *download* state, which starts after the *encrypt* state, is when the recovered images will be downloaded. Thereafter, the process enters the *wait* state again, restarting the state machine. The VHDL source code of the system was synthesized on two FPGA boards, using Vivado version 2019.2 for Artix-7 AC701 and Quartus Prime Lite Edition version 20.1 for Cyclone V, to create a register–transfer level (RTL) architecture. This was followed by the placing and routing implementation process.

In the encryption of the images with each of the three chaotic states of the master, the data of the original RGB and grayscale images were altered with chaos, thereby rendering the original images unintelligible for creating encrypted images. Figure 15b presents the graphical data of the RGB original, encrypted, and decrypted images from chaotic state *x* in the Active-HDL co-simulation. The visualization of the RGB image graphical data on the Handyscope HS3 oscilloscope is shown in Figure 15c. For the grayscale image, the graphical data also from chaotic state *x* are shown in Figure 16b in the Active-HDL co-simulation, whereas the visualization of the graph on the Handyscope HS3 oscilloscope is shown in Figure 16c. Comparing the image graphical data with the Matlab simulations in Figures 15a and 16a, it is seen that the VHDL implementations on the FPGAs were successful as they agree with the Matlab simulations. The images are shown in Figure 17.

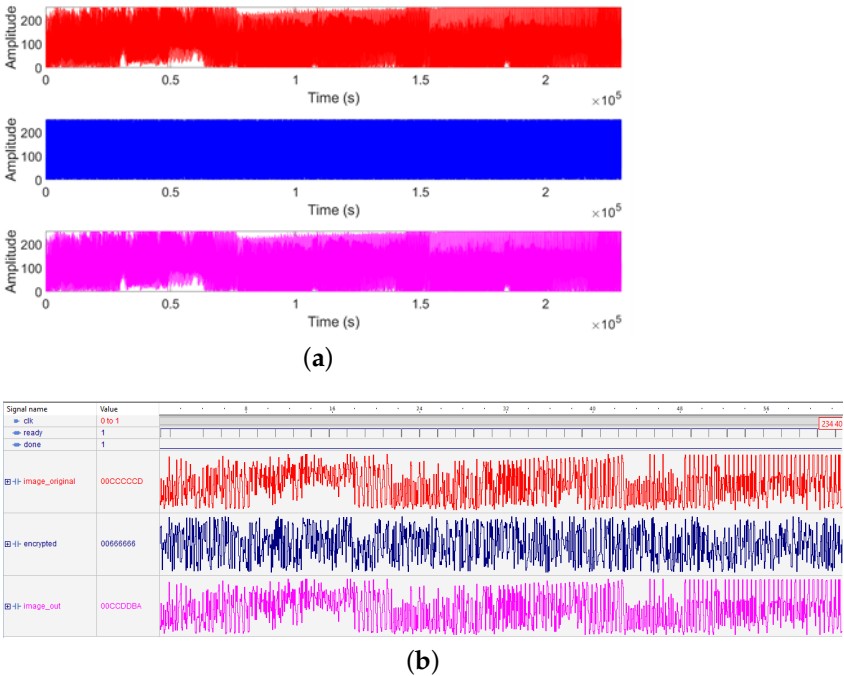

**Figure 15.** *Cont.*

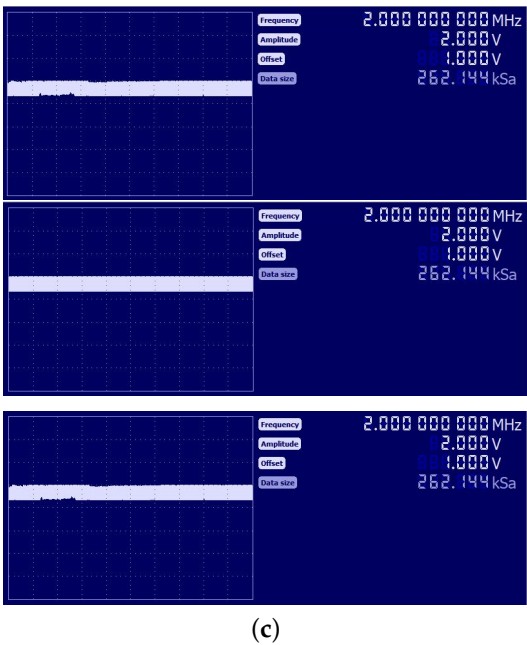

(**c**)

**Figure 15.** The 320 × 240 RGB image transmission based on master system (33) and slave system (35) via chaotic state *x*: (**a**) Matlab simulation showing the graphical data of original (red), encrypted (blue), and decrypted (magenta); (**b**) VHDL simulation in Active-HDL showing the graphical data of original (red), encrypted (blue), and decrypted (magenta); (**c**) original, encrypted, and recovered image signals visualized on TiePie Handyscope HS3 oscilloscope.

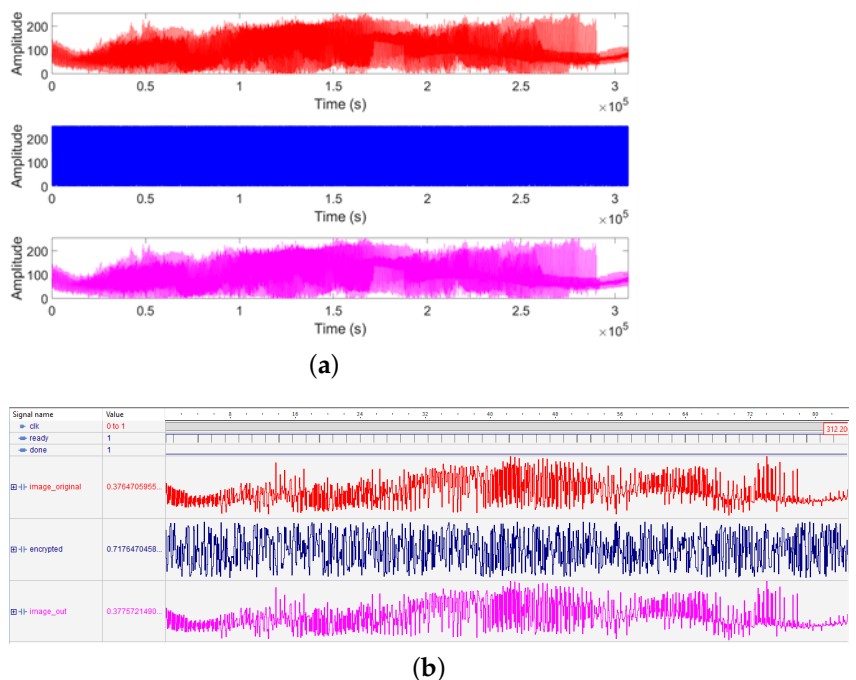

(**a**)

(**b**)

**Figure 16.** *Cont.*

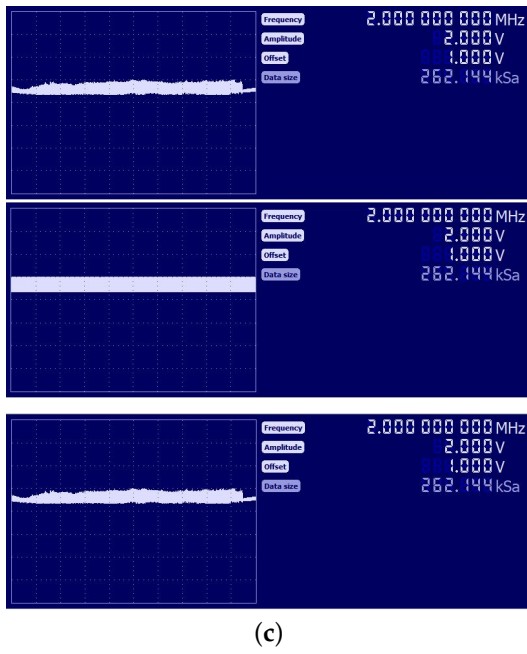

(**c**)

**Figure 16.** The 640 × 480 grayscale image transmission based on master system (33) and slave system (35) via state *x*: (**a**) Matlab simulation showing the graphical data of original (red), encrypted (blue), and decrypted (magenta); (**b**) VHDL simulation in Active-HDL showing the graphical data of original (red), encrypted (blue), and decrypted (magenta); (**c**) original, encrypted, and recovered image signals visualized on TiePie Handyscope HS3 oscilloscope.

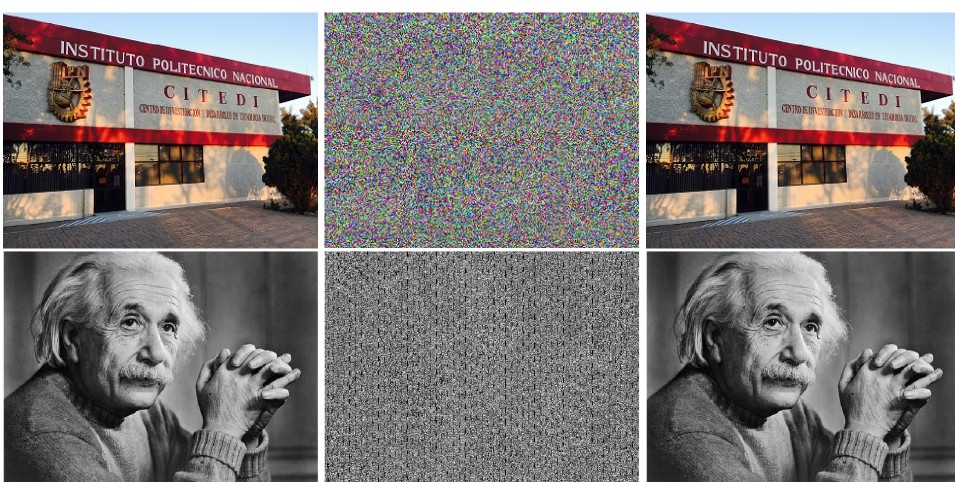

**Figure 17.** Secure image transmission via chaotic state *x* for 320 × 240 RGB and 640 × 480 grayscale images. Left: original; middle: encrypted; right: decrypted.

## 6. Discussion

In this section, we discuss the results of this investigation vis-à-vis the utilization of the logical resources of the FPGA cards, security and statistical analyses, and a comparison with similar works in the literature.

### 6.1. Consumption of Logical Resources

The resources of the Artix-7 AC701 and Cyclone V FPGA boards utilized by the secure image transmission system are presented in Table 1. As seen in the table, the implementations on both boards show that the transmission of the grayscale image utilized more of the logic resources than RGB, as shown by the amount of LUTs consumed. The disparity in the usage of the LUT is attributed to the images having different sizes, which,

in the case of grayscale, consists of 307,200 words, whereas RGB has 230,400 words. The consumed registers, I/O pins, and DSPs are equal because the same codes were used for both images. Overall, a better logic-efficient implementation was achieved in Artix-7 for both RGB and grayscale because the board used fewer LUTs than its Cyclone V counterpart, though the implementation on the latter utilized fewer registers and DSPs than the former. The logic efficiency of the Artix-7 can be attributed to the structure of its LUTs, which are made of configurable logic blocks (CLBs). The Artix-7 CLB consists of two flexible slices with each one having four 6-input LUTs, which can also function as dual 5-input LUTs, eight registers, and two carry chains. This configuration provided a reduction in logic resources and a better performance in Artix 7 than in the Cyclone V Adaptive logic module (ALM), which has an 8-input fracturable LUT, four registers, and two adders.

**Table 1.** Consumption of logic resources by implementation of secure image transmission system on Artix-7 AC701 and Cyclone V.

| FPGA | Resources | Available | RGB Image | | Grayscale Image | |
|---|---|---|---|---|---|---|
| | | | Used | Consumed (%) | Used | Consumed (%) |
| Artix-7 AC701 | LUTs (CLB) | 133,800 | 86,230 | 64 | 92,627 | 69 |
| | Memory LUTs (Kb) | 2888 | 0 | 0 | 0 | 0 |
| | Registers | 267,600 | 2127 | 0.79 | 2127 | 0.79 |
| | I/O Pins | 500 | 195 | 39 | 195 | 39 |
| | Block RAMs (Kb) | 13,140 | 0 | 0 | 0 | 0 |
| | DSPs | 740 | 120 | 16 | 120 | 16 |
| Cyclone V | LUTs (ALM) | 113,560 | 94,019 | 83 | 101,416 | 89 |
| | Memory LUTs (Kb) | 1,717 | 0 | 0 | 0 | 0 |
| | Registers | 454,240 | 420 | 0.09 | 422 | 0.09 |
| | I/O Pins | 616 | 195 | 32 | 195 | 32 |
| | Block RAMs (Kb) | 12,200 | 0 | 0 | 0 | 0 |
| | DSPs | 342 | 96 | 28 | 96 | 28 |

### 6.2. Performance Analysis

The following security and statistical analyses were conducted to examine the performance of the secure image transmission system:

(a) Secret Key Space Analysis: The transmission system presented in this work is sensitive to the parameters, fractional orders, and initial conditions of the underlying chaotic systems, which are the private keys. In this work, the digital block size was 32, out of which 25 bits were dedicated to the fractional part. This means that the precision of each of the parameters $a_1, a_2, a_3, b, c, c_1, c_2, c_3, d, e$, fractional orders $q_1, q_2, q_3$, and initial conditions $x_0, y_0, z_0$ was $10^{-25}$ bits. Therefore, the total number of possibilities in the secret key space was $10^{25 \times 16} = 10^{400}$. This is an enormous size for withstanding a brute force attack by hackers. Table 2 shows a comparison of the secret key space in this work with some others in the literature.

**Table 2.** Secret key space in this work compared to some others in literature.

| This Work | Ref. [56] | Ref. [57] | Ref. [58] |
|---|---|---|---|
| $10^{400}$ | $10^{140} \approx 2^{466}$ | $2^{256}$ | $10^4 \times 2^{208}$ |

(b) Correlation and Jaccard Coefficients Analysis: The closeness of the original image to the encrypted image, as well as the original image to the recovered image, was determined by computing the correlation and Jaccard coefficients in order to compare the results from the three chaotic transmission states, which are presented in Table 3. It should be noted that the correlation coefficient for the RGB image is the average of the R, G, and B channels. Based on the correlation value between the original and encrypted images, the best performance was observed in chaotic transmission state $x$ for the RGB image,

with a value of 0.0511 compared to 0.2178 and 0.0762 for $y$ and $z$, respectively. For the grayscale image, the best result also came from the chaotic transmission state $x$, with a correlation value of 0.0392 compared to 0.0694 and 0.0422 for $y$ and $z$, respectively. As the coefficients are approximately 0, this is an indication that the original image totally differs from the encrypted image; hence, the transmitted images are fully protected. In addition, the coefficient of 1 between the original and recovered images shows that the recovered images are without a loss of quality. The results obtained for the Jaccard metric further confirm the correlation analysis results, whereby the Jaccard similarity coefficient between the original and encrypted images is approximately 0 whereas the dissimilarity (Jaccard distance) is approximately 1. For the original and recovered images, their similarity and dissimilarity are 1 and 0, respectively. The correlation and Jaccard coefficients confirm the graphical data presented in Figures 15 and 16, whereby the graphs of the original and encrypted images show no relativity whereas the graphs of the original and recovered images are the same.

**Table 3.** Correlation coefficients and Jaccard similarity/distance of RGB and grayscale images encryption and decryption in Matlab based on the FOCSS (14).

| Chaotic State | Images | Correlation | | Jaccard Similarity | | Jaccard Distance | |
|---|---|---|---|---|---|---|---|
| | | RGB | Grayscale | RGB | Grayscale | RGB | Grayscale |
| $x$ | Original and encrypted | 0.0511 | 0.0392 | 0.0062 | 0.0059 | 0.9938 | 0.9941 |
| | Original and recovered | 1 | 1 | 1 | 1 | 0 | 0 |
| $y$ | Original and encrypted | 0.2178 | 0.0694 | 0.0069 | 0.0060 | 0.9931 | 0.9940 |
| | Original and recovered | 1 | 1 | 1 | 1 | 0 | 0 |
| $z$ | Original and encrypted | 0.0762 | 0.0422 | 0.0063 | 0.0059 | 0.9937 | 0.9941 |
| | Original and recovered | 1 | 1 | 1 | 1 | 0 | 0 |

(c) Information Entropy: Information entropy is a test index that can be used to analyze the average information content of a random outcome, describing the degree of uncertainty in it. Specifically, the test is used in image processing to measure the distribution of the gray data values in images [59]. Information entropy $E$ is computed according to the next equation:

$$E(G) = -P(g_i)log_2 P(g_i) \tag{38}$$

where $G$ is the image matrix, $g_i$ represents the gray values of the image, and $P(g_i) = Pr(G = g_i)$ is the probability of the $i^{th}$ value of $G$. We present the information entropy analysis of the encrypted RGB and grayscale images in Table 4. An entropy value of 8 means that the image is totally random. In our work, the entropy values of the encrypted images obtained from the three chaotic transmission states are very close to 8; hence, the encryption system is very effective. This also confirms the results of the correlation coefficient and Jaccard similarity index.

**Table 4.** Information entropy $E$ of the encrypted RGB and grayscale images.

| Chaotic State | RGB | Grayscale |
|---|---|---|
| $x$ | 7.9981 | 7.9965 |
| $y$ | 7.9908 | 7.9961 |
| $z$ | 7.9918 | 7.9916 |

(d) Sensitivity to Noise: Here, the effect of noise on the secure image transmission system is investigated. In the context of communication, the term "noise" refers to the contamination that an information in transit may be exposed to over a communication channel [60,61]. In the event that the channel is "noisy", the quality of the recovered information may be affected. Therefore, to investigate the impact of noise, we contaminated the system with noise as follows:

$$\hat{I}_E = I_E + gN \tag{39}$$

where $\hat{I}_E$ is the "noisy" encrypted RGB or grayscale image, $I_E$ is the initial encrypted image, $g$ is the noise intensity coefficient, and $N$ is the introduced noise. In this case, $N$ is an additive white noise of the Gaussian type with zero mean and a variance of 0.01, 0.001, and 0.0001. The results of this investigation are presented in Figure 18 for both RGB and grayscale images, whereas the analysis of the noise impact using the correlation coefficient and signal-to-noise ratio (SNR) is presented in Table 5. The results are for the encrypted images of the transmission state $x$.

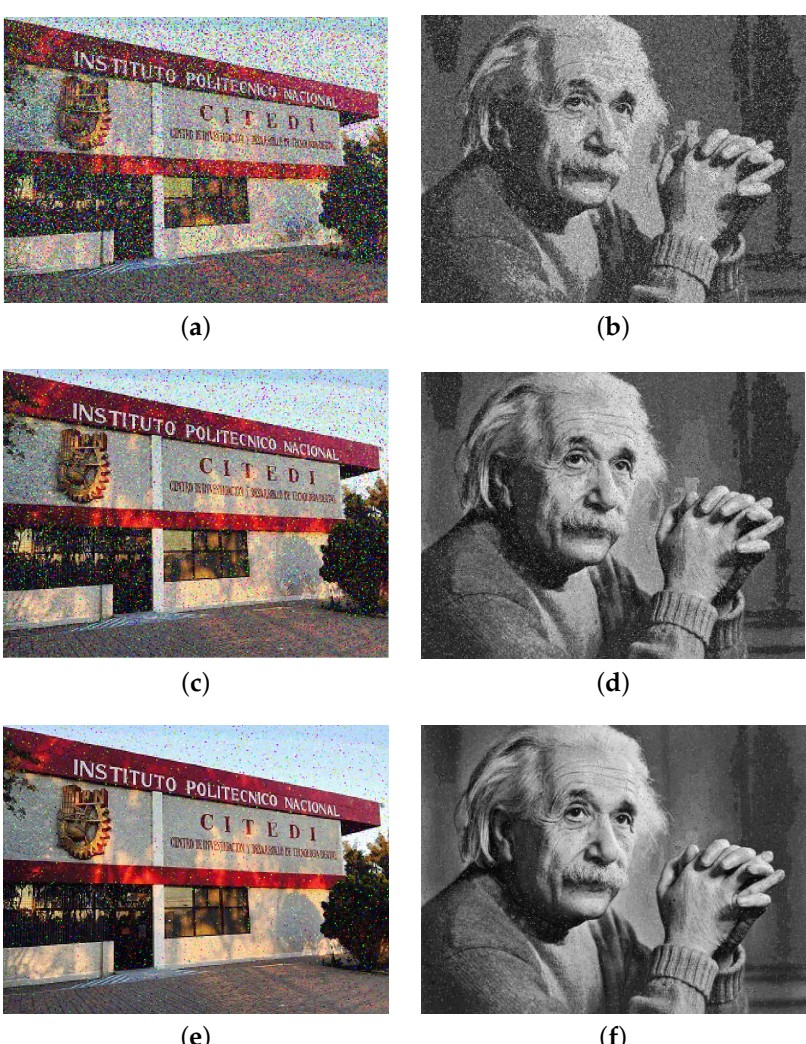

**Figure 18.** RGB and grayscale images recovered after introduction of Gaussian white noise with zero mean: (**a**,**b**) at 0.01 variance, (**c**,**d**) at 0.001 variance, (**e**,**f**) at 0.0001 variance.

The impact of the introduced noise is seen as the recovered images suffered from degradation in response to the different variance of the Guassian noise. The computed SNR, which increases as the noise variance decreases, confirms the negative impact of the introduced noise. The recovery percentage was computed from the correlation coefficient,

and it increases as the variance decreases, with the recovery better in RGB than grayscale. With the recovery percentage shown in the table, the images can be recognized visually. The mitigation of the effect of noise is an investigation for the future.

**Table 5.** Effect of Gaussian white noise with zero mean at different variance values on the recovered images.

| Metric | Variance of 0.01 | | Variance of 0.001 | | Variance of 0.0001 | |
|---|---|---|---|---|---|---|
| | RGB | Grayscale | RGB | Grayscale | RGB | Grayscale |
| Correlation coefficient | 0.7451 | 0.6961 | 0.9071 | 0.8743 | 0.9682 | 0.9556 |
| SNR | 15 dB | 15 dB | 25 dB | 25 dB | 35 dB | 35 dB |
| Percentage recovery | 75% | 70% | 90% | 87% | 96% | 95% |

*6.3. Comparison with Other Works*

Furthermore, the realization of the secure system of image transmission on digital boards was compared with some other related works in the literature as presented in Table 6 based on some parameters such as the chaotic system used, FPGA model and resources, multimedia involved, implementation language, and the numerical approximation method. However, only the data for Artix-7 are shown since it has less logic utilization than Cyclone V.

From the table, it is seen that the implementations in the compared references contain two integer-order systems and one fractional-order system. The reason for this is that, to the best knowledge of the authors, there are very few papers in the literature that apply fractional-order chaotic systems for an image encryption and transmission system. The implementations in our work were found to utilize more FPGA resources than the compared references—64% and 69% for RGB and grayscale, respectively—and this can be attributed to certain factors. First is the complexity of the fractional-order chaotic spherical system, which is the building block of the transmitter and receiver for the image transmission system. The chaotic system contains several nonlinear terms that require enormous computational resources from the FPGAs. In addition, the implementations of the parameter-switching technique and the Grünwald–Letnikov numerical approximation contributed to the complexity of the overall implementation.

The major advantage of our implementation over others is that it is not just an encryption system but also a non-cumbersome communication system, whereby information can be sent from a transmitter to a receiver in a secure manner. Moreover, our implementation is flexible; hence, it can be adapted to encrypt other information, such as texts, videos, and sound.

**Table 6.** Implementation of chaos-based system for image transmission on digital cards in this work versus some related works in literature.

| Parameters | This Work | | Ref. [62] | Ref. [63] | Ref. [64] |
|---|---|---|---|---|---|
| Chaotic system | FOCSS | FOCSS | NMCS | FMCS | NECS |
| FPGA | Artix-7 | Artix-7 | Virtex-6 | Artix-7 | Cyclone IV |
| Image | RGB | Grayscale | RGB | RGB | RGB/Grayscale |
| Image size | $320 \times 240$ | $640 \times 480$ | $256 \times 256$ | $256 \times 256$ | $256 \times 256$ |
| LUTs | 86,230 | 92,627 | 15,978 | 23,929 | 32,983 |
| Registers | 2127 | 2127 | 21,057 | 4599 | 450 |
| I/O pins | 195 | 195 | 16 | N/A | 66 |

**Table 6.** *Cont.*

| Parameters | This Work | | Ref. [62] | Ref. [63] | Ref. [64] |
|---|---|---|---|---|---|
| DSPs | 120 | 120 | 20 | 144 | 84 |
| Language | VHDL | VHDL | VHDL | Verilog | VHDL |
| Numerical approximation | Grünwald–Letnikov | Grünwald–Letnikov | RK4 | Grünwald–Letnikov | RK4 |
| Order | Fractional | Fractional | Integer | Fractional | Integer |

## 7. Conclusions

The work presented in this paper is about implementing the technique of parameter switching to control chaos in fractional-order spherical systems and a chaos-based encryption and transmission system for RGB and grayscale images. Consequently, this investigation led to two major contributions to the current state of the art in fractional-order systems. Firstly, the electronic implementation of the parameter-switching technique to stabilize fractional-order chaotic spherical systems was achieved using the VHDL. The chaos control was demonstrated with two cases of the parameter-switching scheme. Secondly, a methodology was designed and applied to implement an FOCSS-based secure image transmission system on two FPGA boards using the parameter-switching technique as a mechanism for decrypting encrypted information. The secure communication system was based on a synchronized master and slave configuration. The secure image transmission system was used for encrypting, transmitting, and decrypting grayscale and RGB images, where the best result was obtained from the chaotic state $x$ of the fractional-order spherical system as shown by the correlation coefficient of 0.0511 and 0.0392 for the RGB and grayscale images, respectively. The correlation, Jaccard index, and information entropy results show that no similarity exists between the original and encrypted RGB and grayscale images and that both the original and recovered images were the same, meaning that there was no loss of information after the decryption. This work was implemented using VHDL and realized on Xilinx and Intel FPGA boards. The usage of key logical resources of the FPGAs showed that the implementation on the Artix-7 was more logic-efficient.

**Author Contributions:** Conceptualization, J.-C.N.-P., V.-A.A. and E.T.-C.; methodology, J.-C.N.-P., V.-A.A. and E.T.-C.; software, V.-A.A. and Y.S.-I.; validation, J.-C.N.-P., Y.S.-I. and E.T.-C.; formal analysis, J.-C.N.-P., V.-A.A., Y.S.-I. and E.T.-C.; investigation, J.-C.N.-P. and V.-A.A.; resources, J.-C.N.-P. and E.T.-C.; writing—original draft preparation, J.-C.N.-P., V.-A.A. and E.T.-C.; writing—review and editing, J.-C.N.-P., V.-A.A., Y.S.-I. and E.T.-C.; visualization, V.-A.A. and Y.S.-I.; supervision, J.-C.N.-P. and E.T.-C.; project administration, J.-C.N.-P.; funding acquisition, J.-C.N.-P. All authors have read and agreed to the published version of the manuscript.

**Funding:** The authors wish to thank the Instituto Politecnico Nacional for its support provided through the project SIP-20230135. In addition, the authors would like to express their gratitude to the COFAA-IPN for its financial support.

**Institutional Review Board Statement:** Not applicable.

**Informed Consent Statement:** Not applicable.

**Data Availability Statement:** Not applicable

**Conflicts of Interest:** The authors declare no conflict of interest.

## Abbreviations

The following abbreviations are used in this manuscript:

ALM      Adaptive Logic Module
CLB      Configurable Logic Block
DSP      Digital Signal Processor

| FDE | Fractional Differential Equation |
|---|---|
| FMCS | Fractional Memristive Chaotic System |
| FOCSS | Fractional-Order Chaotic Spherical System |
| FONLS | Fractional-Order Nonlinear System |
| FOS | Fractional-Order System |
| FPGA | Field Programmable Gate Array |
| I/O | Input/Output |
| IVP | Initial Value Problem |
| LE | Lyapunov Exponent |
| LUT | Look-up Table |
| MLE | Maximum Lyapunov Exponent |
| N/A | Not Available |
| NECS | No-equilibrium Chaotic System |
| NMCS | New Multi-scroll Chua's System |
| RGB | Red, Green, Blue |
| UPO | Unstable Periodic Orbit |
| VHDL | VHSIC Hardware Description Language |

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
