# Peer review of "FPGA Implementation of Parameter-Switching Scheme to Stabilize Chaos in Fractional Spherical Systems and Usage in Secure Image Transmission"

_fractalfract, doi:10.3390/fractalfract7060440_

Round 1

Reviewer 1 Report

The paper proposed a robust methodology to stabilize chaos in a fraction-order and spherical chaotic system. The proposed methodology is demonstrated by theoretical analysis, numerical simulation, hardware implementation, and engineering application. The obtained results show that it is a good methodology. The paper is well written. The idea is novel and interesting. The analysis is sufficient. However, I suggest that the authors address the following problems.

(1)   Scientific writing should be improved. For example, the abstract is too confusing. I suggest the authors write in order of purpose, method, result, and conclusion.

(2)   In the introduction, the motivation and innovation of this paper are not outstanding enough. Furthermore, some recent related references are missing, such as chaos-based image encryption application in “An extremely simple multiwing chaotic system: dynamics analysis, encryption application, and hardware implementation”

(3)   In Fig.2, the authors give the bifurcation diagram related to parameter c3. However, the corresponding Lyapunov exponents are missing. Please add it.

(4)   The texts in some pictures such as Fig.3, Fig.10, and Fig.13 are not clear.

(5)   Please describe the numerical method as well as the algorithm used clearly.

(6)   In section 6, other test indexes including information entropy, key sensitivity, and data loss and noise attack should be analyzed.

(7)   There are too few references. Some recent important papers such as https://doi.org/10.53391/mmnsa.2022.01.004; https://doi.org/10.1109/TCSII.2022.3212394; https://doi.org/10.1063/1.5096645; https://doi.org/10.1016/j.chaos.2021.111332; https://doi.org/10.3390/math11030767; https://doi.org/10.1016/j.neunet.2020.02.008 should be cited.

Author Response

REPLY TO REVIEWERS

Title: FPGA-Realization of Parameter-Switching Scheme to Stabilize Chaos in Fractional and Spherical Systems for Image Encryption

Authors: Vincent Ademola Adeyemi, Esteban Tlelo Cuautle, Yuma Sandoval Ibarra, and Jose Cruz Nuñez Perez

April 18, 2023.

We would like to thank the reviewers for the time and effort they have spent on reviewing our manuscript and providing us with the invaluable comments and suggestions. We carefully read the comments and suggestions of the reviewers and revised the manuscript accordingly.

On the following pages, we outline point-by-point the responses to the comments and suggestions of the reviewers. In the attached manuscript, the changes are highlighted in colored text in red color means MODIFYING.

It is our hope that the revised version of our manuscript will be received favorably, and we look forward to hearing from you soon.

Thank you.

REVIEWER 1

The paper proposed a robust methodology to stabilize chaos in a fraction-order and spherical chaotic system. The proposed methodology is demonstrated by theoretical analysis, numerical simulation, hardware implementation, and engineering application. The obtained results show that it is a good methodology. The paper is well written. The idea is novel and interesting. The analysis is sufficient. However, I suggest that the authors address the following problems.

(1) Scientific writing should be improved. For example, the abstract is too confusing. I suggest the authors write in order of purpose, method, result, and conclusion.

Response:

As advised and following your recommendation, we have re-written the abstract.

(2) In the introduction, the motivation and innovation of this paper are not outstanding enough. Furthermore, some recent related references are missing, such as chaos-based image encryption application in “An extremely simple multiwing chaotic system: dynamics analysis, encryption application, and hardware implementation”

Response:

We have improved the introduction by highlighting the motivation and innovation in this work. This can be found in Lines 93 – 101. Also, the suggested reference has been cited as [39].

(3) In Fig.2, the authors give the bifurcation diagram related to parameter c3. However, the corresponding Lyapunov exponents are missing. Please add it.

Response:

That Lyapunov exponents of the bifurcation plot has been included.

(4) The texts in some pictures such as Fig.3, Fig.10, and Fig.13 are not clear.

Response:

The texts in Fig. 3 have been made more legible. For Figs. 10 and 13, the text size has been fixed by the application that generated the figures. However, the text can be seen clearly by zooming in the document.

(5) Please describe the numerical method as well as the algorithm used clearly.

Response:

The description of the numerical method is contained in Section 2.1, which was formerly titled Fractional-Order Chaotic Systems. Also, we have included the encryption algorithm in Fig. 13.

(6) In section 6, other test indexes including information entropy, key sensitivity, and data loss and noise attack should be analyzed.

Response:

Thank you for your suggestion. We have included information entropy in Subsection 6.2 (c), while data loss and noise attack are shown in Subsection 6.2 (d).

(7) There are too few references. Some recent important papers such as

https://doi.org/10.53391/mmnsa.2022.01.004;

https://doi.org/10.1109/TCSII.2022.3212394;        

https://doi.org/10.1063/1.5096645;  

https://doi.org/10.1016/j.chaos.2021.111332;

https://doi.org/10.3390/math11030767        

https://doi.org/10.1016/j.neunet.2020.02.008

should be cited.

Response:

The suggested references have been cited in the paper as follows:

https://doi.org/10.53391/mmnsa.2022.01.004 - [28].

https://doi.org/10.1109/TCSII.2022.3212394 - [23].    

https://doi.org/10.1063/1.5096645 - [40].        

https://doi.org/10.1016/j.chaos.2021.111332 - [22].     

https://doi.org/10.3390/math11030767 - [42].  

https://doi.org/10.1016/j.neunet.2020.02.008 - [41].

Other recent papers that have also been cited are [29], [30], [43], [44], [45], [46], [47].

Acknowledgment

We would like to profoundly thank the anonymous reviewers for their time and effort on reviewing our manuscript, and for their comments and invaluable suggestions which helped us much to improve our work.

Reviewer 2 Report

In this paper, a robust method for stabilizing fractional order and spherical chaotic systems is presented, and numerical and electronic implementations are given. The Hamiltonian method with observer-based approach was applied to synchronize chaotic spherical systems of fractional order. A methodology was designed and applied to implement secure image transmission system on two FPGA cards using the parameter-switching technique as a mechanism for decrypting an encrypted information. The numerical results verify the reliability of the image transmission system. However, the following problems remain in this paper. 

1. The main idea of the title is not clear and needs to be revised.

2. Abstract presentation is redundant and needs to be streamlined to make the logic clearer.

3. Introduction has too much representation of chaos and lacks description of image encryption and fractional order.

4. Adjust the parameter interval of Figure 1 to make the three spectral lines clearer. 

5. Subgraphs in Figure 3 and Figure 5 are not labeled. 

6. In the fifth section, the image encryption algorithm is not clear, should add a flow chart, etc.    

7. In the sixth section, subtitles should be added appropriately to make the logical structure clearer.

Author Response

REPLY TO REVIEWERS

Title: FPGA-Realization of Parameter-Switching Scheme to Stabilize Chaos in Fractional and Spherical Systems for Image Encryption

Authors: Vincent Ademola Adeyemi, Esteban Tlelo Cuautle, Yuma Sandoval Ibarra, and Jose Cruz Nuñez Perez

April 18, 2023.

We would like to thank the reviewers for the time and effort they have spent on reviewing our manuscript and providing us with the invaluable comments and suggestions. We carefully read the comments and suggestions of the reviewers and revised the manuscript accordingly.

On the following pages, we outline point-by-point the responses to the comments and suggestions of the reviewers. In the attached manuscript, the changes are highlighted in colored text in red color means MODIFYING.

It is our hope that the revised version of our manuscript will be received favorably, and we look forward to hearing from you soon.

Thank you.

 REVIEWER 2

In this paper, a robust method for stabilizing fractional order and spherical chaotic systems is presented, and numerical and electronic implementations are given. The Hamiltonian method with observer-based approach was applied to synchronize chaotic spherical systems of fractional order. A methodology was designed and applied to implement secure image transmission system on two FPGA cards using the parameter-switching technique as a mechanism for decrypting an encrypted information. The numerical results verify the reliability of the image transmission system. However, the following problems remain in this paper.

(1) The main idea of the title is not clear and needs to be revised.

Response:

The title has been revised as follows:

FPGA Implementation of Parameter-Switching Scheme to Stabilize Chaos in Fractional Spherical Systems and Usage in Secure Image Transmission

(2) Abstract presentation is redundant and needs to be streamlined to make the logic clearer.

Response:

We have re-composed the abstract as recommended.     

(3) Introduction has too much representation of chaos and lacks description of image encryption and fractional order.

Response:

The introduction section has been updated as suggested. We have reduced the discussion on chaos and included fractional order in Lines 20 – 46 and image encryption is contained in Lines 81 – 92.

(4) Adjust the parameter interval of Figure 1 to make the three spectral lines clearer.

Response:

We have adjusted the parameter interval of Figure 1 as advised.

(5) Subgraphs in Figure 3 and Figure 5 are not labeled.

Response:

The subgraphs in Figures 3 and 5 have been labeled.

(6) In the fifth section, the image encryption algorithm is not clear, should add a flow chart, etc.

Response:

As advised, we have included the flowchart of the image encryption in Figure 13 and mentioned in Line 432.

(7) In the sixth section, subtitles should be added appropriately to make the logical structure clearer.

Response:

We have organized Section 6 into three subsections. The subsections are:

6.1. Consumption of Logical Resources

6.2. Performance Analysis

(a) Secret Key Space Analysis:

(b) Correlation and Jaccard Coefficients Analysis:

(c) Information Entropy:

(d) Sensitivity to Noise:

6.3. Comparison with Other Works

Acknowledgment

We would like to profoundly thank the anonymous reviewers for their time and effort on reviewing our manuscript, and for their comments and invaluable suggestions which helped us much to improve our work.

Reviewer 3 Report

The paper presents some aspects that should be improved befor the acceptance:

1)The RTL system implemented on the FPGA is not explained in deep.

It is not clear how the entity is made inside in terms of the circuit. If some algebraic processing is performed (ex addition, multiplication etc) authors have to explain the hardware modules involved in this computation. For example how the sine function of (33) is performed in hardware?

2)comparison table number 4 should be more commented underlining the advantage of the proposed method. In addition, also the throughput of the systems should be provided. Usually in an FPGA implementation comparisons are done in terms of hardware resources, timing and power consumption

3)"The hardware design was based on 32-bit number format which is composed of 1 bit for sign, 6 bits for integer component, and 25 bits for the fractional part. The floating-point computations were made possible by two IEEE libraries, namely f ixed_pkg and f ixed_ f loat_types, while communication of  data among entities was performed with the s f fixed type"

This sentence is not clear. the format 1-6-25 seems to be a fixed point format so what is the meaning of the sentence "The floating-point computations ....."

Author Response

REPLY TO REVIEWERS

Title: FPGA-Realization of Parameter-Switching Scheme to Stabilize Chaos in Fractional and Spherical Systems for Image Encryption

Authors: Vincent Ademola Adeyemi, Esteban Tlelo Cuautle, Yuma Sandoval Ibarra, and Jose Cruz Nuñez Perez

April 18, 2023.

We would like to thank the reviewers for the time and effort they have spent on reviewing our manuscript and providing us with the invaluable comments and suggestions. We carefully read the comments and suggestions of the reviewers and revised the manuscript accordingly.

On the following pages, we outline point-by-point the responses to the comments and suggestions of the reviewers. In the attached manuscript, the changes are highlighted in colored text in red color means MODIFYING.

It is our hope that the revised version of our manuscript will be received favorably, and we look forward to hearing from you soon.

Thank you.

REVIEWER 3

The paper presents some aspects that should be improved before the acceptance:

(1) The RTL system implemented on the FPGA is not explained in deep.

It is not clear how the entity is made inside in terms of the circuit. If some algebraic processing is performed (ex addition, multiplication etc) authors have to explain the hardware modules involved in this computation. For example how the sine function of (33) is performed in hardware?

Response:

We considered the option of reducing the computational intensity involved in the fractional-order model used in this work. One way we achieved this is by avoiding the computation of the hyperbolic tangent function in the model on the FPGA, which consists of the sine and exponential functions. Instead, we created a LUT for the hyperbolic tangent function. We have included this explanation in Lines 363 – 368.

(2) Comparison table number 4 should be more commented underlining the advantage of the proposed method. In addition, also the throughput of the systems should be provided. Usually in an FPGA implementation comparisons are done in terms of hardware resources, timing and power consumption.

Response:

We have improved the discussion on Table 6 (formerly Table 4) by including the advantage of our implementation. This can be found in Lines 584 – 588. Our focus in the comparison is the hardware resources utilized by the implementations. Moreover, the compared references only presented the hardware resources utilization in their work, hence, our decision to base our comparison on the resource consumption.

(3) "The hardware design was based on 32-bit number format which is composed of 1 bit for sign, 6 bits for integer component, and 25 bits for the fractional part. The floating-point computations were made possible by two IEEE libraries, namely fixed_pkg and fixed_ f loat_types, while communication of data among entities was performed with the s f fixed type"

This sentence is not clear. The format 1-6-25 seems to be a fixed point format so what is the meaning of the sentence "The floating-point computations ....."

Response:

We thank you for making this observation. We meant to write fixed-point computation. The error has been corrected as shown in Line 361.

Acknowledgment

We would like to profoundly thank the anonymous reviewers for their time and effort on reviewing our manuscript, and for their comments and invaluable suggestions which helped us much to improve our work.

Round 2

Reviewer 1 Report

The revised paoper has addressed all my questions, and can be published.

Author Response

We would like to thank the reviewer for the time and effort he has spent on reviewing our manuscript and providing us with the invaluable comments and suggestions. 

Reviewer 2 Report

In this paper, a robust method for stabilizing fractional order and spherical chaotic systems is presented, and numerical and electronic implementations are given. The Hamiltonian method with observer-based approach was applied to synchronize chaotic spherical systems of fractional order. A methodology was designed and applied to implement secure image transmission system on two FPGA cards using the parameter switching technique as a mechanism for decrypting an encrypted information. The numerical results verify the reliability of the image transmission system. However, the following problems remain in this paper.

1.Introduction should be appropriately streamlined.

2.There are formatting errors in the text, such as the citation mark "[1?]" in the first sentence of the introduction.

3.The operations in the algorithm flowchart should be described briefly.

4.Conclusion should be revised appropriately to make it more logical.

Author Response

REPLY TO REVIEWERS

 Title: FPGA Implementation of Parameter-Switching Scheme to Stabilize Chaos in Fractional Spherical Systems and Usage in Secure Image Transmission

Authors: Vincent Ademola Adeyemi, Esteban Tlelo Cuautle, Yuma Sandoval Ibarra, and Jose Cruz Nuñez Perez

REV 2.0

May 1, 2023.

Reviewer 2

In this paper, a robust method for stabilizing fractional order and spherical chaotic systems is presented, and numerical and electronic implementations are given. The Hamiltonian method with observer-based approach was applied to synchronize chaotic spherical systems of fractional order. A methodology was designed and applied to implement secure image transmission system on two FPGA cards using the parameter switching technique as a mechanism for decrypting an encrypted information. The numerical results verify the reliability of the image transmission system. However, the following problems remain in this paper.

  1. Introduction should be appropriately streamlined.

Response:

The introduction has been streamlined.

  1. There are formatting errors in the text, such as the citation mark "[1?]" in the first sentence of the introduction.

Response:

Reference [1] is part of the newly included references in the introduction.

  1. The operations in the algorithm flowchart should be described briefly.

Response:

The brief description of the operations in the algorithm’s flowchart has been provided in lines 425-431.

  1. Conclusion should be revised appropriately to make it more logical.

Response:

The conclusion has been revised.
